# Machine Learning for Variance Reduction in Online Experiments

**Yongyi Guo**
Department of Operations Research and Financial Engineering
Princeton University
Princeton, NJ 08544
yongyig@princeton.edu

**Dominic Coey**
Facebook
1 Hacker Way, Menlo Park, CA 94025
coey@fb.com

**Mikael Konutgan**
Facebook
1 Hacker Way, Menlo Park, CA 94025
kmikael@fb.com

**Wenting Li**
Facebook
1 Hacker Way, Menlo Park, CA 94025
wentingli@fb.com

**Chris Schoener**
Facebook
1 Hacker Way, Menlo Park, CA 94025
chrissc@fb.com

**Matt Goldman**
Facebook
1 Hacker Way, Menlo Park, CA 94025
mattgoldman@fb.com

## Abstract

We consider the problem of variance reduction in randomized controlled trials, through the use of covariates correlated with the outcome but independent of the treatment. We propose a machine learning regression-adjusted treatment effect estimator, which we call MLRATE. MLRATE uses machine learning predictors of the outcome to reduce estimator variance. It employs cross-fitting to avoid over-fitting biases, and we prove consistency and asymptotic normality under general conditions. MLRATE is robust to poor predictions from the machine learning step: if the predictions are uncorrelated with the outcomes, the estimator performs asymptotically no worse than the standard difference-in-means estimator, while if predictions are highly correlated with outcomes, the efficiency gains are large. In A/A tests, for a set of 48 outcome metrics commonly monitored in Facebook experiments the estimator has over 70% lower variance than the simple difference-in-means estimator, and about 19% lower variance than the common univariate procedure which adjusts only for pre-experiment values of the outcome.

## 1 Introduction

While sample sizes are typically larger for online experiments than traditional field experiments, the desired minimum detectable effect sizes may be small, and the outcome variables of interest may be heavy-tailed. Even with quite large samples, statistical power may be low. Variance reduction methods play a key role in these settings, allowing for precise inferences with less data [12, 32, 38].

35th Conference on Neural Information Processing Systems (NeurIPS 2021).

One common technique involves "adjusting" the simple difference-in-means estimator to account for covariate imbalances between the test and control groups [12, 22], with the magnitude of the adjustment depending on both the magnitude of those imbalances, and the correlation between the covariates and the outcome of interest. If covariates are highly correlated with the outcome, then the treatment effect estimator's variance will decrease substantially.

Performing this adjustment procedure with the pre-experiment values of the outcome variable itself as the covariate can greatly reduce confidence interval (CI) width, if the outcome exhibits high autocorrelation. A natural question is how to adjust for multiple covariates, which may have a complicated nonlinear relationship with the outcome variable. Using many covariates in a machine learning (ML) model, it may be possible to develop a proxy highly correlated with the outcome variable and hence generate further variance reduction gains. This raises both statistical and scalability issues, however, as it is unclear how traditional justifications for linear regression adjustment with a fixed number of covariates translate to the case of general, potentially very complex ML methods, and it may not be scalable to generate new predictions every time an experiment's results are queried.

This paper makes three contributions. First, we propose an easy-to-implement and practical estimator which can take full advantage of ML methods to perform regression adjustment across a potentially large number of covariates, and derive its asymptotic properties. We name the procedure MLRATE, for "machine learning regression-adjusted treatment effect estimator". MLRATE uses cross-fitting (e.g. [3, 9, 25]), which simplifies the asymptotic analysis and guarantees that the "naive" CIs which do not correct for the ML estimation step are asymptotically valid. We also ensure robustness of the estimator to poor quality predictions from the ML stage, by including those ML predictions as a covariate in a subsequent linear regression step. Our approach is agnostic or model-free in two key respects—we do not assume that the ML model converges to the truth, and in common with [22], in the subsequent linear regression step we do not assume that the true conditional mean is linear. Second, we demonstrate that the method works well for online experiments in practice. Across a variety of metrics, the estimator reduces variance in A/A tests by around 19% on average relative to regression adjustment for pre-experiment outcomes only. Some metrics see variance reduction of 50% or more. Variance reduction of this magnitude can amount to the difference between experimentation being infeasibly noisy and being practically useful. Third, we sketch how the computational considerations involved in implementing MLRATE at scale can be surmounted.

## 2  Outcome prediction for variance reduction

### 2.1  Setup & motivation

The data consist of a vector of covariates $X$, an outcome variable $Y$, and a binary treatment indicator $T$. The treatment is assigned randomly and independently of the covariates. For observations $i = 1, 2, \ldots, N$, the vector $(Y_i, X_i, T_i)$ is drawn iid from a distribution $P$. To motivate our main estimator and illustrate some of the central ideas, first consider a "difference-in-difference"-style estimator, where we train an ML model $g(X)$ predicting $Y$ from $X$, and then compute the difference between the test and control group averages of $Y - g(X)$. If we treat the estimated ML model $g$ as non-random and ignore its dependence on the sample, the resulting estimator has the same expectation as the usual difference-in-means estimator where we compute the difference between the test and control group averages of $Y$. This is because $g(X)$ and $T$ are independent, and hence $E[Y - g(X) \mid T = 1] - E[Y - g(X) \mid T = 0] = E[Y \mid T = 1] - E[Y \mid T = 0]$. Furthermore, if $g(X)$ is a good predictor of $Y$, then $Var(Y)$ will exceed $Var(Y - g(X))$, and the difference-in-difference estimator based on averages of $Y - g(X)$ will be lower variance than the difference-in-means estimator based on averages of $Y$.

MLRATE differs in two main respects from the heuristic argument above. First, instead of directly subtracting the ML predictions $g(X)$ from the outcome $Y$, we include them as a regressor in a subsequent linear regression step. This guarantees robustness of the estimator to poor, even asymptotically inconsistent predictions: regardless of how bad the outcome predictions from the ML step are, MLRATE has an asymptotic variance no larger than the difference-in-means estimator. Second, we use cross-fitting to estimate the predictive models, so that the predictions for every observation are generated by a model trained only on other observations. This allows us to control the randomness in the ML function ignored in the argument above. We derive the asymptotic distribution of this regression-adjusted estimator, and show that the usual, "naive" CIs for the average treatment

effect (ATE), which ignore the randomness generated by estimating the predictive models, are in fact asymptotically valid. Thus asymptotically the ML step can only increase precision, and introduces no extra complications in computing CIs.

## 2.2 Related work

Our work is closely related to the large literature on semiparametric statistics and econometrics, in which low-dimensional parameters of interest are estimated in the presence of high-dimensional nuisance parameters [6, 29, 23, 33, 21, 37]. A common approach in this literature appeals to Donsker conditions and empirical process theory to control the randomness generated by the estimation error in the nuisance function [1, 36, 35, 18]. This approach is less appealing in this context, as it would greatly restrict the kind of ML methods that could be used for the prediction step [9], and hence the variance reduction attainable. The idea of instead using sample-splitting in semiparametric problems—estimating nuisance parameters on one subset of data and evaluating them on another—dates back at least to [6], with subsequent contributions by [30, 19, 7], among others. More recent applications of this idea, also referred to as "cross-fitting", include [9, 3, 4, 40]. This paper is especially similar in spirit to "double machine-learning" [9], which combines sample-splitting with the use of Neyman-orthogonal scores, which have the property of being insensitive to small errors in estimating the nuisance function. Although the results of [9] do not directly carry over to our setting, we use similar arguments to establish our results. A second strand of related literature concerns "agnostic" regression adjustment, which delivers consistent estimates of the average treatment estimate even when the regression model is misspecified [39, 14, 22, 16, 11]. The procedure described in [11] is particularly relevant, as it shares the same structure of first estimating nonlinear models and then calibrating them in a linear regression step, although in contrast to their work we use sample-splitting to allow for a very general class of nonlinear ML models. A third strand of the literature considers improving estimator precision in the context of large-scale online experiments [8, 12, 10, 38].

Relative to these literatures, our contribution is to describe an estimator which i) delivers substantial variance reduction when good ML predictors of the outcome variable are available, and ii) performs well in the presence of poor-quality predictions, even allowing for predictive models which *never* converge to the truth with infinite data. In particular, MLRATE is guaranteed to never perform worse, asymptotically, than the difference-in-means estimator, even with arbitrarily poor predictions. In contrast to low-dimensional regression adjustment, we give formal statistical guarantees on inference even when complex ML models are used to predict outcomes; in contrast to double-ML, as applied to randomized experiments, our proposed estimator need not be semiparametrically efficient, but allows for inconsistent estimates of the nuisance parameters. Finally, this methodology is practical and computationally efficient enough to be deployed at large scale, and we show with Facebook data that MLRATE can deliver substantial additional variance reduction beyond the existing state-of-the-art commonly used in practice, of linear regression adjustment for pre-experiment covariates [12, 38].

## 2.3 Estimation and inference with MLRATE

The linear regression-adjusted estimator of the ATE is the OLS estimate of $\alpha_1$ in the regression

$$Y_i = \alpha_0 + \alpha_1 T_i + \alpha_2 X_i + \alpha_3 T_i(X_i - \overline{X}) + \epsilon_i, \tag{1}$$

where $\overline{X}$ is the average of $X_i$ over all $i$. The covariates $X_i$ may be multivariate, but are of fixed dimension that does not grow with the sample size. The analysis in [22] establishes that the OLS estimator for $\widehat{\alpha}_1$ is a consistent and asymptotically normal estimator of the ATE $E[Y \mid T = 1] - E[Y \mid T = 0]$, and the robust, Huber-White standard errors are asymptotically valid. In contrast to this setting, we wish to capture complex interactions and nonlinearities in the relationship between the outcome and covariates, and allow for a vector of covariates with dimension potentially increasing

with the sample size. To this end, we propose the following procedure. We assume throughout that $N$ is evenly divisible by $K$, to simplify notation.

---

**Algorithm 1:** Estimation and inference with MLRATE

---

**Input:** Data $(Y_i, X_i, T_i)_{i=1}^N$ split uniformly at random into $K$ equal-sized splits. $I_k :=$ index set of the $k$-th split and $I_k^c := \{1, 2, \ldots, N\} \setminus I_k, \forall k$. $\mathcal{M}$, a supervised learning algorithm.
**Result:** ML regression-adjusted ATE estimator $\widehat{\alpha}_1$, asymptotic variance $\widehat{\sigma}^2$.
**for** $k \leftarrow 1$ **to** $K$ **do**
  | Generate the function $\widehat{g}_k$ predicting $Y_i$ given $X_i$, by applying $\mathcal{M}$ to the sample $(Y_i, X_i)_{i \in I_k^c}$.
**end**
Compute $\overline{g} = \frac{1}{N} \sum_i \widehat{g}_{k(i)}(X_i)$, where $k(i) :=$ the split index containing observation $i$;
Compute $\widehat{\alpha}_1$ as the OLS estimator for $\alpha_1$ in
  $Y_i = \alpha_0 + \alpha_1 T_i + \alpha_2 \widehat{g}_{k(i)}(X_i) + \alpha_3 T_i \left( \widehat{g}_{k(i)}(X_i) - \overline{g} \right) + \varepsilon_i$;
Compute $\widehat{\sigma}^2$ according to (11).

---

Section 2.4 proves the statistical validity of this estimation and inference procedure. Note that if the cross-fitted, random functions $\{\widehat{g}_k\}_{k=1}^K$ were replaced by a single, fixed function $g$, MLRATE reduces to the standard linear regression-adjusted estimator. Instead the relation between the covariates $X_i$ and the outcome $Y_i$ is itself estimated from the data. Intuitively this should help with variance reduction, as the estimated proxy $\widehat{g}_{k(i)}(X_i)$ may be highly correlated with $Y_i$, but with the challenge that the dependence of the $\widehat{g}_k$'s on the data complicates the analysis of the statistical properties of the treatment effect estimator $\widehat{\alpha}_1$. Our main technical result assuages this concern, showing that the asymptotic distribution of $\widehat{\alpha}_1$ is not impacted and thus it is a consistent, asymptotically normal estimator of the ATE. CIs with level $100(1-a)$ percent are given in the usual way by $\widehat{\alpha}_1 \pm \Phi^{-1}(1-a/2)\widehat{\sigma}/\sqrt{N}$, where $\Phi$ is the CDF of the standard normal distribution.

**Remark 1.** The chief purpose of cross-fitting is to avoid bias from overfitting. With sufficiently flexible ML models, in-sample predictions would be close to the outcomes $Y_i$. The linear regression step would then amount to adjusting for the outcome variable itself, which is correlated with the treatment, and this may introduce severe attenuation bias into estimates of the treatment effect. By generating predictions only on out-of-sample data, we ensure the adjustment covariate is independent of the treatment.

**Remark 2.** In online experiments, only a subset of users are typically assigned to any given experiment. To maximize training data and minimize compute costs, an equally valid variation on the above is to perform the cross-fitting ML step once, using data from all users, whereas the linear regression step must occur separately for every experiment of interest, using only the users in that experiment.

**Remark 3.** Alternatively, one may estimate a single ML model *entirely* on pre-experiment data. For an experiment starting at time $t$, we may train a model predicting time $t-1$ outcomes from time $t-2$ covariates, and then use that model to predict time $t$ outcomes from time $t-1$ covariates. Those model predictions can then be treated as any other covariate, as they are entirely a function of pre-experiment data, and the results of [22] apply. Although simpler, this approach suffers from the drawbacks that it requires some history of the outcome metric to exist even pre-experiment, and that the predictive model may perform worse if the relationship between covariates and outcomes changes over time.

**Remark 4.** The choice of $K$ does not affect the asymptotic distribution of the estimator, although it may matter in finite samples. As [9] note, in cross-fitting applications involving estimating high-dimensional nuisance functions with small samples, larger values of $K$ (e.g. $K = 4$ or 5) may perform better. Much larger values of $K$ may be unattractive, however, given diminishing returns in model performance and the extra compute cost. In the simulations and empirical examples in Section 3 we show that good performance is achievable even with the low computation choice of $K = 2$.

We now sketch the main technical result. Beyond standard regularity conditions, the main assumption is that for each split $k$, the estimated functions $\widehat{g}_k$ converge to some $g_0$ in the sense that $\int [\widehat{g}_k(X) - g_0(X)]^4 dP \rightarrow_p 0$. This condition is quite weak in two aspects: On the one hand, it only requires consistency of the $\widehat{g}_k$ to $g_0$, and not convergence at a particular rate. Such consistency results are available for many common ML algorithms, including random forests (see [2] and references therein), gradient boosted decision trees [5], deep feedforward neural nets [13], and regularized linear regression in some asymptotic regimes [20]. On the other hand, we allow the ML modelling step

to be misspecified and inconsistent: there is no requirement that $g_0(X) = E[Y \mid X]$, although a poorly-specified ML model may limit the variance reduction obtained. Allowing for inconsistent estimators is an especially important advantage in the presence of high-dimensional covariates, as in such settings there is no general guarantee that ML estimators will be consistent if the number of covariates grows faster than $\log(N)$, due to the curse of dimensionality [31].

To make explicit the dependence on the $\widehat{g}_k$'s, we denote MLRATE by $\widehat{\alpha}_1(\{\widehat{g}_k\}_{k=1}^K)$. We denote by $\widehat{\alpha}_1(g_0)$ the linear regression adjustment estimator as in (1) where we adjust for the covariate $g_0(X_i)$. This latter estimator is infeasible as $g_0$ is unknown, but we prove in Theorem 1 below that the two estimators are asymptotically equivalent, i.e. $\sqrt{N}[\widehat{\alpha}_1(\{\widehat{g}_k\}_{k=1}^K) - \widehat{\alpha}_1(g_0)] \to_p 0$. Deriving the asymptotic distribution of $\widehat{\alpha}_1(g_0)$ is straightforward, and from this equivalence we conclude that $\widehat{\alpha}_1(\{\widehat{g}_k\}_{k=1}^K)$ shares the same asymptotic distribution.

## 2.4 The asymptotic behavior of MLRATE

Define the covariate vector as a function of an arbitrary (possibly random) function $g$, $Z(g) = (1, T, g(X), Tg(X))^\top$, and define $Z_i(g) = (1, T_i, g(X_i), T_i g(X_i))^\top$ for $i = 1, \ldots, N$. We adopt the notation $Pg = \int g dP$.[1] In what follows, matrix norms refer to the operator norm, and $\lambda_{min}(M)$ denotes the minimum eigenvalue of the symmetric matrix $M$. Recall that given the assumption of equally-sized splits, $N = Kn$. All proofs are in the Appendix.

**Assumption 1.** *i)* $p \in (0, 1)$. *ii) For all* $k = 1, 2, \ldots, K$, *the estimated functions* $\widehat{g}_k$ *belong to a vector space of functions* $\mathcal{G}$ *with probability one, with* $\mathcal{G}$ *satisfying* $\sup_{g \in \mathcal{G}} P[|g|^{4+\delta}] < \infty$ *for some* $\delta > 0$. *iii)* $P[Y^4] < \infty$. *iv) For each* $k = 1, 2, \ldots, K$, $\widehat{g}_k$ *converges to some function* $g_0 \in \mathcal{G}$ *in the sense that* $\int [\widehat{g}_k(X) - g_0(X)]^4 dP \to_p 0$. *v)* $\inf_{g \in \mathcal{G}} Var(g(X)) > 0$.

Condition i) is a standard assumption in randomized controlled trials, while conditions ii) and iii) are standard boundedness requirements. Condition iv) is the convergence assumption discussed above. Condition v) can be motivated with reference to the scientific question at hand: restricting attention only to adjustment functions which exhibit nontrivial variation with respect to the value of the covariate is unlikely to hurt the amount of variance reduction achieved.

The following proposition ensures that the inverse of $P[Z(g)Z(g)^\top]$ exists for all $g$.

**Proposition 1.** *Given Assumption 1,* $\inf_{g \in \mathcal{G}} \lambda_{min}(P[Z(g)Z(g)^\top]) > 0$.

Define

$$\widehat{\beta}(\{\widehat{g}_k\}_{k=1}^K) = \left[ \frac{1}{N} \sum_k \sum_{i \in I_k} Z_i(\widehat{g}_k) Z_i(\widehat{g}_k)^\top \right]^{-1} \left[ \frac{1}{N} \sum_k \sum_{i \in I_k} Z_i(\widehat{g}_k) Y_i \right], \tag{2}$$

and

$$\beta(\{\widehat{g}_k\}_{k=1}^K) = \left[ \frac{1}{K} \sum_k P[Z(\widehat{g}_k) Z(\widehat{g}_k)^\top] \right]^{-1} \left[ \frac{1}{K} \sum_k P[Z(\widehat{g}_k)Y] \right]. \tag{3}$$

These are the sample and population OLS coefficients, from the regression of $Y_i$ on $Z_i(\widehat{g}_{k(i)})$. We also define the corresponding quantities for the limiting function $g_0$,

$$\widehat{\beta}(g_0) = \left[ \frac{1}{N} \sum_i Z_i(g_0) Z_i(g_0)^\top \right]^{-1} \left[ \frac{1}{N} \sum_i Z_i(g_0) Y_i \right], \tag{4}$$

and

$$\beta(g_0) = \left[ P[Z(g_0)Z(g_0)^\top] \right]^{-1} P[Z(g_0)Y]. \tag{5}$$

The key intermediate step in deriving the asymptotic distribution of MLRATE is the following result, which states that the distribution of $\widehat{\beta}(\{\widehat{g}_k\}_{k=1}^K)$, centered around the random variable $\beta(\{\widehat{g}_k\}_{k=1}^K)$, is asymptotically equivalent to that of $\widehat{\beta}(g_0)$, centered around $\beta(g_0)$.

---

[1]If the input function $\widehat{g}$ is random, the quantity $P\widehat{g}$ is also a random variable. If it is a deterministic function $g$, then $Pg$ is the same as the expectation $E[g(X)]$.

**Proposition 2.** *Under Assumption 1,*

$$\sqrt{N} \left\| [\widehat{\beta}(\{\widehat{g}_k\}_{k=1}^K) - \beta(\{\widehat{g}_k\}_{k=1}^K)] - [\widehat{\beta}(g_0) - \beta(g_0)] \right\| \to_p 0.$$

Having established Proposition 2, we turn to the limiting distribution of MLRATE. This is not quite immediate: the estimator is defined as the coefficient on $T_i$ in the regression of $Y_i$ on a constant, $T_i$, $\widehat{g}_{k(i)}(X_i)$, and $T_i(\widehat{g}_{k(i)} - \overline{g})$. By contrast, Proposition 2 concerns the regression of $Y_i$ on a constant, $T_i$, $\widehat{g}_{k(i)}(X_i)$, and $T_i\widehat{g}_{k(i)}$. To conclude the argument we write MLRATE in terms of the coefficients from the latter regression, and apply Proposition 2. MLRATE, $\widehat{\alpha}_1(\{\widehat{g}_k\}_{k=1}^K)$, can be written as

$$\widehat{\alpha}_1(\{\widehat{g}_k\}_{k=1}^K) = \widehat{\beta_1}(\{\widehat{g}_k\}_{k=1}^K) + \widehat{\beta_3}(\{\widehat{g}_k\}_{k=1}^K)\frac{1}{N}\sum_i \widehat{g}_{k(i)}(X_i). \tag{6}$$

We also define $\widehat{\alpha}_1(g_0)$, the corresponding quantity using the unknown $g_0$:

$$\widehat{\alpha}_1(g_0) = \widehat{\beta_1}(g_0) + \widehat{\beta_3}(g_0)\frac{1}{N}\sum_i g_0(X_i). \tag{7}$$

The ATE $\alpha_1$ satisfies

$$\alpha_1 = \beta_1(g_0) + \beta_3(g_0)Pg_0 = \beta_1(\{\widehat{g}_k\}_{k=1}^K) + \beta_3(\{\widehat{g}_k\}_{k=1}^K)\frac{1}{K}\sum_{k=1}^K P\widehat{g}_k. \tag{8}$$

Here $\beta_i(\{\widehat{g}_k\}_{k=1}^K)$ denotes the $i$-th entry of $\beta(\{\widehat{g}_k\}_{k=1}^K)$; $\beta_i(g_0)$ and $\widehat{\beta}_i(\{\widehat{g}_k\}_{k=1}^K)$ are similarly defined. To see why (8) hold, note that $\alpha_1 = \beta_1(g) + \beta_3(g)Pg$ holds for *any* function $g$: this is essentially a restatement of the observation that regardless of the particular covariate we adjust for, the regression-adjusted estimator will still be consistent for the ATE [39, 34]. This argument resembles the idea of Neyman orthogonality ([26, 9]), where the estimate of the parameter of interest is not heavily influenced by an undesirable estimate of the nuisance function.

We now state our main theorem, which asserts that the randomness from the ML function fitting step in MLRATE does not affect its asymptotic distribution.

**Theorem 1.** *Under Assumption 1,*

$$\sqrt{N}\left[\widehat{\alpha}_1(\{\widehat{g}_k\}_{k=1}^K) - \widehat{\alpha}_1(g_0)\right] \to_p 0.$$

*Consequently $\sqrt{N}\left[\widehat{\alpha}_1(\{\widehat{g}_k\}_{k=1}^K) - \alpha_1\right]$ and $\sqrt{N}\left[\widehat{\alpha}_1(g_0) - \alpha_1\right]$ are asymptotically equivalent.*

Given Theorem 1, the problem of finding the asymptotic distribution of MLRATE reduces to finding the asymptotic distribution of $\widehat{\alpha}_1(g_0)$. The latter, summarized in the following proposition, can be established by standard asymptotic arguments, and is already known in the literature.[2] Define $p = E(T)$, $\sigma_g^2 = Var(g_0(X_i))$, $\sigma_{Y_C}^2 = Var(Y_i \mid T_i = 0)$, and $\sigma_{Y_T}^2 = Var(Y_i \mid T_i = 1)$. For notational convenience, below we use $\beta_{0,i}$ to denote $\beta_i(g_0)$ for each $i$.

**Proposition 3.** *If $E(g_0(X)^2) < \infty$, $E(Y^2) < \infty$, and $0 < p < 1$, then $\sqrt{N}\left[\widehat{\alpha}_1(g_0) - \alpha_1\right] \rightsquigarrow \mathcal{N}(0, \sigma^2)$, where*

$$\sigma^2 = \frac{\sigma_{Y_C}^2}{1-p} + \frac{\sigma_{Y_T}^2}{p} - \frac{\sigma_g^2}{p(1-p)}\left[\beta_{0,2}p + (\beta_{0,2} + \beta_{0,3})(1-p)\right]^2. \tag{9}$$

Putting together the previous results, we arrive at the asymptotic distribution for MLRATE, which is asymptotically normal and centered around the ATE $\alpha_1$.

**Corollary 1.** *Under Assumption 1, $\sqrt{N}\left[\widehat{\alpha}_1(\{\widehat{g}_k\}_{k=1}^K) - \alpha_1\right] \rightsquigarrow \mathcal{N}(0, \sigma^2)$, where*

$$\sigma^2 = \frac{\sigma_{Y_C}^2}{1-p} + \frac{\sigma_{Y_T}^2}{p} - \frac{\sigma_g^2}{p(1-p)}\left[\beta_{0,2}p + (\beta_{0,2} + \beta_{0,3})(1-p)\right]^2. \tag{10}$$

---

[2]See, for example, equation (10) in [39]. Note that there is a small typo in that display: it should read $\Sigma_2 = \frac{1}{1-\delta}\sigma_{22}^{(0)} + \frac{1}{\delta}\sigma_{22}^{(1)} - \frac{1}{\delta(1-\delta)\sigma_{11}}\left\{(1-\delta)\sigma_{12}^{(1)} + \delta\sigma_{12}^{(0)}\right\}^2$ instead of $\Sigma_2 = \frac{1}{1-\delta}\sigma_{22}^{(0)} + \frac{1}{\delta}\sigma_{22}^{(1)} - \frac{1}{\delta(1-\delta)\sigma_{11}}\left\{(1-\delta)\sigma_{12}^{(1)} + \delta\sigma_{22}^{(0)}\right\}^2$.

It follows directly from this corollary that the asymptotic variance of MLRATE is smaller than variance of the simple difference-in-means estimator by the amount

$$\frac{\sigma_g^2}{p(1-p)}\left[\beta_{0,2}p + (\beta_{0,2} + \beta_{0,3})(1-p)\right]^2 \geq 0.$$

Thus ML regression adjustment, like ordinary linear regression adjustment [39, 22], cannot reduce asymptotic precision. For some intuition about the determinants of variance reduction, consider the special case where $\beta_{0,3} = 0$ (i.e. the slope of the best-fitting linear relationship between $Y$ and $g_0(X)$ does not vary from test to control groups), and $\sigma_{Y_C}^2 = \sigma_{Y_T}^2$. The unadjusted, difference-in-means estimator has asymptotic variance $\sigma_{Y_C}^2/(1-p) + \sigma_{Y_T}^2/p$. The relative efficiency of the adjusted estimator, $\sigma^2/[\sigma_{Y_C}^2/(1-p) + \sigma_{Y_T}^2/p]$, equals $1 - Corr(Y, g_0(X))^2$. If $Corr(Y, g_0(X)) = 0.5$, regression adjustment shrinks CIs by $1 - \sqrt{1 - 0.5^2} = 13.4\%$; with a correlation of 0.8, they are 40% smaller.

The following proposition shows that the sample analog of (10) is a consistent estimator of the asymptotic variance, and it can thus be used to construct asymptotically valid CIs.

**Proposition 4.** *Let $\widehat{\sigma}^2$ be the sample analog of $\sigma^2$, that is,*

$$\widehat{\sigma}^2 = \frac{\widehat{Var}(Y_i \mid T_i = 0)}{1 - \widehat{p}} + \frac{\widehat{Var}(Y_i \mid T_i = 1)}{\widehat{p}} \tag{11}$$
$$- \frac{\widehat{Var}(\widehat{g}_{k(i)}(X_i))}{\widehat{p}(1-\widehat{p})}\left[\widehat{\beta}_2(\{\widehat{g}_k\}_{k=1}^K)\widehat{p} + \left(\widehat{\beta}_2(\{\widehat{g}_k\}_{k=1}^K) + \widehat{\beta}_3(\{\widehat{g}_k\}_{k=1}^K)\right)(1-\widehat{p})\right]^2,$$

*where $\widehat{p} = \sum_i T_i/N$. Under Assumption 1, $\widehat{\sigma}^2 \to_p \sigma^2$.*

## 3 Simulations & empirical results

We now validate MLRATE in practice, on both simulated data, and real Facebook user data. These two validation exercises serve complementary purposes: simulations allow us to verify that the CIs' empirical coverage is indeed close to their nominal coverage for the data generating process of our choice, while the Facebook data gives an indication of the magnitude of variance reduction that can be expected in practice. All computation is done on an internal cluster, on a standard 64GB ram machine.

Our simulated data generating process has $N = 10{,}000$ iid observations and 100 covariates distributed as $X_i \sim \mathcal{N}(0, I_{100 \times 100})$. The outcome variable is $Y_i = b(X_i) + T_i \tau(X_i) + u_i$, where $b(\cdot)$ is the Friedman function $b(X_i) = 10 \sin(\pi X_{i1} X_{i2}) + 20(X_{i3} - 0.5)^2 + 10 X_{i4} + 5 X_{i5}$ and the treatment effect function is $\tau(X_i) = X_{i1} + \log(1 + \exp(X_{i2}))$ [15, 27]. The treatment indicator is $T_i \sim \text{Bernoulli}(0.5)$, and the error term is $u_i \sim \mathcal{N}(0, 25^2)$. Treatment is independent of covariates and the error term, and the error term is independent of the covariates. This data generating process involves non-trivial complexity, with nonlinearities and interactions in the baseline outcome, many extraneous covariates that do not affect outcomes, and heterogeneous treatment effects correlated with some covariates. We find the ATE by Monte Carlo integration, and compute the average number of times the MLRATE CIs contain this ATE, over 10,000 simulation repetitions, as well as 95% CIs for this coverage percentage. Both in these simulations and the subsequent analysis of Facebook data, we choose gradient boosted regression trees (GBDT) and elastic net regression as two examples of ML prediction procedures in MLRATE, with scikit-learn's implementation [28]. Moreover, we choose $K = 2$ splits for cross-fitting.

Table 1 shows the simulation results. "CI Coverage" displays the average coverage percentage rate over the 10,000 simulations, and the CI width for these estimated coverage rates. "Relative CI Width" displays the CI width for each method divided by the simple difference-in-means CI width ("Unadjusted"), averaged over the 10,000 simulations. Empirical coverage is close to the nominal coverage for all three estimators, with the CIs for empirical coverage including the nominal rate. Both the GBDT and elastic net versions of MLRATE demonstrate efficiency gains over the difference-in-means estimator. As might be expected given the highly nonlinear dependence of the outcomes on covariates, GBDT performs substantially better than the linear, elastic net model:

Table 1: CI coverage and variance reduction results of MLRATE-GBDT and MLRATE-Elastic Net on complex nonlinear simulated data. "CI Coverage" displays the average coverage percentage rate over 10,000 simulations, and the CI width for these estimated coverage rates. "Relative CI Width" displays the CI width for each method divided by the simple difference-in-means CI width ("Unadjusted"), averaged over the 10,000 simulations.

|  | MLRATE-GBDT | MLRATE-Elastic Net | Unadjusted |
|---|---|---|---|
| CI Coverage (%) | $95.18 \pm 0.42$ | $95.34 \pm 0.41$ | $94.88 \pm 0.43$ |
| Relative CI Width | 0.62 | 0.86 | 1.00 |

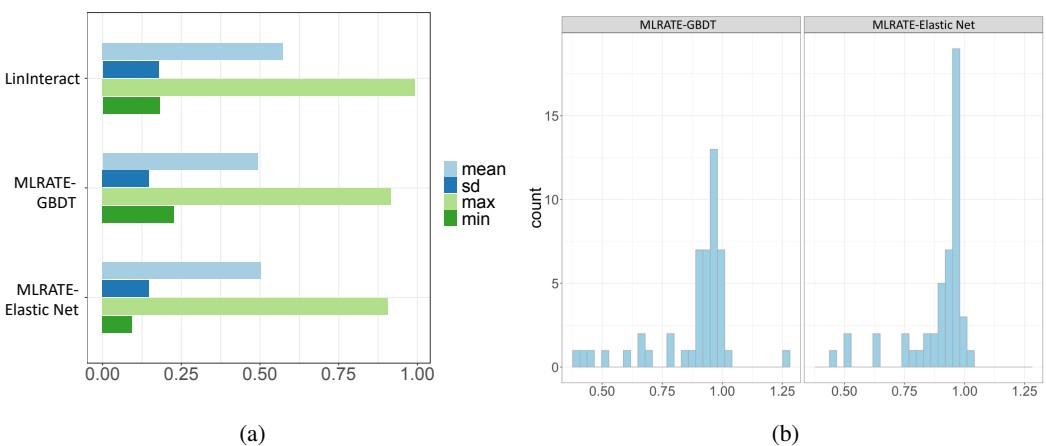

(a)                                                      (b)

Figure 1: Variance reduction results on 48 real metrics used in online experiments run by Facebook. Confidence intervals (CI) are calculated by sampling $\sim 400,000$ observations for each metric. (a) Mean, standard deviation, maximum and minimum value of relative variance of LinInteract, MLRATE-GBDT and MLRATE-Elastic Net compared to the difference-in-means estimator among the metrics. (b) Distribution of CI width of MLRATE-GBDT and MLRATE-Elastic Net relative to LinInteract, by metric.

on average across simulations, the MLRATE-GBDT CIs are 62% the width of the unadjusted CIs, whereas the analogous figure for the elastic net CIs is 86%. Also of interest is the comparison to the semiparametric efficiency bound [24, 17], which can be calculated explicitly for this data generating process: despite the fact that MLRATE is agnostic, and does not assume consistency of the ML procedure employed, the MLRATE-GBDT CIs are only 11.3% wider than those implied by the semiparametric efficiency bound.

The variance reduction numbers above are of course dependent on the particular data generating process specified in the simulation. To get a better sense of the magnitudes of variance reduction one might expect in practice, we evaluate the estimator on 48 real metrics used in online experiments run by Facebook, capturing a broad range of the most commonly consulted user engagement and app performance measurements. We focus on A/A tests in this evaluation rather than A/B tests run in production. This is because the true effect is unknown in the latter, which makes it impossible to evaluate the coverage properties of the CI. Because treatments in online experiments are typically subtle and are unlikely to greatly change the relationship between outcomes and covariates, the magnitude of variance reduction will likely be very similar in A/B tests.

For each outcome metric, we select a random sample of approximately 400,000 users, and simulate an A/A test by assigning a treatment indicator for each user, drawn from a Bernoulli(0.5) distribution. The features used in the ML model vary for each metric and consist of the pre-experiment values of the metric, as well as the pre-experiment values of other metrics that have been grouped together as belonging to the same product area. There are between 20 and 100 other such metrics, with the exact number depending on the outcome metric in question. Outcome values are calculated as the sum of the daily values over a period of one week, and the features values are calculated as the sum of the daily values over the three weeks leading up to the experiment start date.

For each metric, we calculate variances and CI width for four estimators of the ATE: The difference-in-means estimator; A univariate linear regression adjustment procedure of equation (1) where the only covariate $X_i$ is the pre-experiment value of the outcome metric $Y_i$ (for simplicity, we denote it by 'LinInteract'); And MLRATE-GBDT/Elastic Net with all available pre-experiment metrics used as features.

Figure 1a shows that LinInteract substantially outperforms the simple difference-in-means estimator, and MLRATE delivers additional gains still. Unlike in the simulated data generating process above, MLRATE-GBDT and MLRATE-Elastic Net perform similarly. The variance reduction relative to the difference-in-means estimator is 72 - 74% on average across metrics, and relative to LinInteract is 19%. The corresponding figures for reduction of the average CI width are 50 - 51%, and 11 - 12%, respectively. Alternatively, to achieve the same precision as the MLRATE-GBDT estimator, the difference-in-means estimator would require sample sizes on average 5.44 times as large on average across metrics and the univariate procedure would require sample sizes 1.56 times as large.

Figure 1b displays the metric-level distribution of CI widths relative to the univariate adjustment case. There is substantial heterogeneity in performance across metrics: for some, ML regression adjustment delivers only quite modest gains relative to univariate adjustment, while for others, it drastically shrinks CIs. This is natural given the variety of metrics in the analysis: some, especially binary or discrete outcomes, may benefit more from more sophisticated predictive modelling, whereas for others simple linear models may perform well. For some metrics, CIs are shrunk by half or more, which may be the difference between experimentation for those metrics being practical and not. As in the simulations, the coverage rates for ML regression adjusted CIs for these metrics are close to the nominal level. For the metric experiencing the largest variance reduction gains from MLRATE—where one might be the most concerned with coverage—we find an average coverage rate of 94.90% over 10,000 simulated A/A tests, where each simulated A/A test is carried out on a 10% subsample drawn at random with replacement from the initial user dataset.

We remark that we design our evaluation to give a realistic sense of the potential variance reduction gains that can be attained with minimal effort and common software implementations of standard ML algorithms. In fact, the supervised learning models we use in this analysis–GBDT and elastic net regression–are deliberately simple, and the training data sample sizes of around 400,000 observations are not especially large by the standards of online A/B tests. The input features to the models are not heavily preprocessed: they are typically raw logged metric values, as opposed to, say, embeddings generated by a prior ML layer. Moreover, as already mentioned in remark 4, we always choose the number of splits $K = 2$ instead of treating it as a hyperparameter and tuning for better performance. We expect that with more sophisticated supervised learning techniques (e.g. deep, recurrent neural networks with transfer learning across metrics), larger datasets, and better choice of $K$ through cross-validation, the precision gains could be considerably greater still.[3]

In the simulations and the empirical study above, the dimension of the covariates is not large compared to the sample size. However, our algorithm applies equally to the high-dimensional regime. In many high-dimensional applications, Assumption 1 can be easily satisfied, and our theory fully extends to this case.

## 4    Implementation

The key guiding principle for selecting features for the ML model is that we can use any variables independent of treatment assignment. Thus any variable extracted before the experiment start is eligible. This simple rule facilitates collaboration with engineering and data science partners familiar with forecasting: they can freely apply their domain expertise to engineer features and build predictive models for specific metrics, without concerns about statistical validity as long as the cross-fitting step in MLRATE is enforced.

The ML step in MLRATE means that the analyst can err on the side of inclusivity in deciding what features to use, as irrelevant features will tend to be omitted from the fitted model. In contrast to [12], for example, we are automatically learning the one 'feature' $(\widehat{g}_{k(i)}(X_i))$ that has the best predictive

---

[3]In Figure 1b, one metric in the GBDT case has substantially *larger* variance than the univariate adjustment case, indicating that the default GBDT fit performs quite poorly on this sample. Larger sample sizes or more customized ML modeling will have the benefit of attenuating such anomalies.

power instead of restricting ourselves to a particular pre-experiment feature, thus allowing for greater overall variance reduction. Moreover, this method is highly scalable as the ML step does not need to be performed once per experiment. Once predictions have been generated for a given metric, they can be used to improve precision for all experiments starting after the period used for feature construction.

For real-world applications, the linear regression step in MLRATE, which ensures non-inferiority relative to the difference-in-means estimator, is an important safeguard. There may be no guarantee in practice that the predictive models produced by modeling teams will always be well-calibrated, and without the linear regression layer this non-inferiority guarantee need not hold.

Finally, we note that an additional "censoring" step may be useful when the metric has substantial mass close to zero, reducing computation cost without significantly affecting estimation accuracy. After training the models $\{\widehat{g}_k\}_{k=1,2,...,K}$, instead of regression adjustment using $\{\widehat{g}_{k(i)}(X_i)\}$, define $\widehat{g}_\tau(X_i) = \mathcal{T}(\widehat{g}_{k(i)}(X_i), \tau)$ for some pre-determined threshold $\tau$, where $\mathcal{T}$ is the hard-thresholding operator $\mathcal{T}(u, \tau) = u1_{\{u \geq \tau\}}$. Then one can perform regression adjustment with $\widehat{g}_\tau(X_i)$ in place of $\widehat{g}_{k(i)}(X_i)$, with the same statistical theory applying. Small values of $\tau$ will cause small efficiency losses, but can greatly reduce the computation cost on the linear regression when $N$ is large.[4]

## 5    Conclusion

MLRATE is a scalable methodology that allows ML algorithms to be used for variance reduction, while still giving formal statistical guarantees on consistency and CI coverage. Of particular practical importance is the methodology's robustness to the ML algorithm used, both in the sense that the ML algorithm used need not be consistent for the truth, and in the sense that no matter how bad the ML predictions are, MLRATE has asymptotic variance no larger than the difference-in-means estimator. Our application to Facebook data demonstrates variance reduction gains using pre-experiment covariates and even simple predictive algorithms. We expect that more sophisticated predictive algorithms, and incorporating other covariates into this framework–for example, generating user covariates by synthetic-control inspired strategies that incorporate contemporaneous data on outcomes for individuals *outside* the experiment–could lead to more substantial efficiency gains still.

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
