# Supplementary Material for
# Machine Learning for Variance Reduction in Online Experiments

**Yongyi Guo**
Department of Operations Research and Financial Engineering
Princeton University
Princeton, NJ 08544
`yongyig@princeton.edu`

**Dominic Coey**
Facebook
1 Hacker Way, Menlo Park, CA 94025
`coey@fb.com`

**Mikael Konutgan**
Facebook
1 Hacker Way, Menlo Park, CA 94025
`kmikael@fb.com`

**Wenting Li**
Facebook
1 Hacker Way, Menlo Park, CA 94025
`wentingli@fb.com`

**Chris Schoener**
Facebook
1 Hacker Way, Menlo Park, CA 94025
`chrissc@fb.com`

**Matt Goldman**
Facebook
1 Hacker Way, Menlo Park, CA 94025
`mattgoldman@fb.com`

In this supplementary material, we provide the proof of all theoretical results stated in the paper.

## 1 Proof of Proposition 1

For any (deterministic) $g \in \mathcal{G}$, we have

$$P[Z(g)Z(g)^{\top}] = M_1(g) \otimes M_2,$$

where $\otimes$ denotes the Kronecker product,

$$M_1(g) = \begin{pmatrix} 1 & Eg(X) \\ Eg(X) & Eg(X)^2 \end{pmatrix}, \quad M_2 = \begin{pmatrix} 1 & p \\ p & p \end{pmatrix}.$$

Therefore, any eigenvalue of $P[Z(g)Z(g)^{\top}]$ is the product of one eigenvalue of $M_1(g)$ and one eigenvalue of $M_2$. It's easy to verify from Assumption 1 that all eigenvalues of $M_1(g)$ and $M_2$ are nonnegative and bounded. Thus, we only need to show $\inf_{g \in \mathcal{G}} \lambda_{min}(M_1(g)) > 0$, $\lambda_{min}(M_2) > 0$.

Through some calculations, one can find out that

$$\lambda_{min}(M_1(g)) = \frac{1}{2}\Big\{(Eg(X)^2 + 1) - \sqrt{(Eg(X)^2 + 1)^2 - 4Var(g(X))}\Big\}$$

$$= \frac{2Var(g(X))}{(Eg(X)^2 + 1) + \sqrt{(Eg(X)^2 + 1)^2 - 4Var(g(X))}} \geq \frac{Var(g(X))}{Eg(X)^2 + 1},$$

35th Conference on Neural Information Processing Systems (NeurIPS 2021), Sydney, Australia.

which leads to

$$\inf_{g \in \mathcal{G}} \lambda_{min}(M_1(g)) \geq \frac{\inf_{g \in \mathcal{G}} Var(g(X))}{\sup_{g \in \mathcal{G}} Eg(X)^2 + 1} > 0.$$

On the other hand, $\lambda_{min}(M_2) > 0$ can be deduced from $p \in (0,1)$. By combining the above two inequalities, we conclude the proof.

## 2  Proof of Proposition 2

For compactness we may write the random variables $Z(\widehat{g}_k)$ as $\widehat{Z}_k$ and $Z(g_0)$ as $Z$. Similarly for any observation $i$ we write $Z_i(\widehat{g}_k)$ as $\widehat{Z}_{k,i}$ and $Z_i(g_0)$ as $Z_i$. We are only interested in convergence in probability, so we can assume that the inverse matrices in the definition of $\widehat{\beta}(\{\widehat{g}_k\}_{k=1}^K)$ and $\widehat{\beta}(g_0)$ exist, as this happens with probability approaching 1 according to Lemma 2. We have $\widehat{\beta}(\{\widehat{g}_k\}_{k=1}^K) - \beta(\{\widehat{g}_k\}_{k=1}^K) = A + B$, where

$$A = \underbrace{\left[\left[\frac{1}{N}\sum_k \sum_{i \in I_k} \widehat{Z}_{k,i}\widehat{Z}_{k,i}^\top\right]^{-1} - \left[\frac{1}{K}\sum_k P[\widehat{Z}_k\widehat{Z}_k^\top]\right]^{-1}\right]}_{F_0} \cdot \left[\frac{1}{N}\sum_k \sum_{i \in I_k} \widehat{Z}_{k,i}Y_i\right],$$

and

$$B = \left[\frac{1}{K}\sum_k P[\widehat{Z}_k\widehat{Z}_k^\top]\right]^{-1} \underbrace{\left[\frac{1}{N}\sum_k \sum_{i \in I_k}[\widehat{Z}_{k,i}Y_i - P[\widehat{Z}_kY]]\right]}_{G_0}.$$

Similarly, $\widehat{\beta}(g_0) - \beta(g_0) = C + D$, where

$$C = \underbrace{\left[\left[\frac{1}{N}\sum_i Z_i Z_i^\top\right]^{-1} - [P[ZZ^\top]]^{-1}\right]}_{F_1} \left[\frac{1}{N}\sum_i Z_i Y_i\right]$$

and

$$D = [P[ZZ^\top]]^{-1} \underbrace{\left[\frac{1}{N}\sum_i[Z_iY_i - P[ZY]]\right]}_{G_1}.$$

We can write $[\widehat{\beta}(\{\widehat{g}_k\}_{k=1}^K) - \beta(\{\widehat{g}_k\}_{k=1}^K)] - [\widehat{\beta}(g_0) - \beta(g_0)] = A - C + B - D$. We show that $\sqrt{N}\|A - C\| \to_p 0$ and $\sqrt{N}\|B - D\| \to_p 0$. From the definitions of $F_0$ and $F_1$ above, we have $A - C = [F_0 - F_1]\left[\frac{1}{N}\sum_k \sum_{i \in I_k} \widehat{Z}_{k,i}Y_i\right] + F_1\left[\frac{1}{N}\sum_k \sum_{i \in I_k}(\widehat{Z}_{k,i} - Z_i)Y_i\right]$. If

1. $\left\|\sqrt{N}[F_0 - F_1]\right\| = o_p(1)$

2. $\left\|\frac{1}{N}\sum_k \sum_{i \in I_k} \widehat{Z}_{k,i}Y_i\right\| = O_p(1)$

3. $\left\|\sqrt{N}F_1\right\| = O_p(1)$

4. $\left\|\frac{1}{N}\sum_k \sum_{i \in I_k}(\widehat{Z}_{k,i} - Z_i)Y_i\right\| = o_p(1)$,

then $\sqrt{N}\|A - C\| = o_p(1)$ as desired. Similarly we write $B - D$ as $B - D = \left[\left[\frac{1}{K}\sum_k P[\widehat{Z}_k\widehat{Z}_k^\top]\right]^{-1} - [P[ZZ^\top]]^{-1}\right]G_0 + [P[ZZ^\top]]^{-1}[G_0 - G_1]$. If

5. $\left\|\left[\frac{1}{K}\sum_k P[\widehat{Z}_k\widehat{Z}_k^\top]\right]^{-1} - [P[ZZ^\top]]^{-1}\right\| = o_p(1)$

6. $\left\| \sqrt{N} G_0 \right\| = O_p(1)$

7. $\left\| P[ZZ^\top]^{-1} \right\| = O_p(1)$

8. $\left\| \sqrt{N}[G_0 - G_1] \right\| = o_p(1)$

then $\sqrt{N} \|B - D\| = o_p(1)$ as desired. We complete the proof in 8 steps by showing statements 1 - 8 above.

**Step 1.** We apply Lemma 3 by letting $M_{1n} = \frac{1}{N} \sum_k \sum_{i \in I_k} \widehat{Z}_{k,i} \widehat{Z}_{k,i}^\top$, $B_n = M_{2n} = P[ZZ^\top]$, $A_n = M_{3n} = \frac{1}{K} \sum_k P[\widehat{Z}_k \widehat{Z}_k^\top]$, $M_{4n} = \frac{1}{N} \sum_k \sum_{i \in I_k} Z_i Z_i^\top$. Consequently, Step 1 amounts to verifying the conditions of Lemma 3. In fact, these conditions are guaranteed by Lemma 1 as well as the following fact: For each $k = 1, \dots, K$,

$$\left\| \frac{1}{\sqrt{n}} \sum_{i \in I_k} \left[ \widehat{Z}_{k,i} \widehat{Z}_{k,i}^\top - P[\widehat{Z}_k \widehat{Z}_k^\top] - Z_i Z_i^\top + P[ZZ^\top] \right] \right\| \to_p 0. \tag{1}$$

We now prove (1). Define $W_{k,i} = \widehat{Z}_{k,i} \widehat{Z}_{k,i}^\top - P[\widehat{Z}_k \widehat{Z}_k^\top] - Z_i Z_i^\top + P[ZZ^\top]$, and note that conditional on the data in $I_k^c$, the function $\widehat{g}_k$ is non-random, and the $W_{k,i}$ are mean zero matrices, uncorrelated across observations in $I_k$. With slight abuse of notation, we use $E[\cdot \mid I_k^c]$ to denote expectations conditional on the observations with indices belonging to the set $I_k^c$. For any $k = 1, 2, \dots, K$,

$$E\left[ \left\| \frac{1}{\sqrt{n}} \sum_{i \in I_k} W_{k,i} \right\|^2 \bigg| I_k^c \right] = \frac{1}{n} E\left[ \operatorname{tr} \left( \sum_{i,j \in I_k} W_{k,i}^\top W_{k,j} \right) \bigg| I_k^c \right] \tag{2}$$

$$= \frac{1}{n} E\left[ \sum_{i \in I_k} \operatorname{tr} \left( W_{k,i}^\top W_{k,i} \right) \bigg| I_k^c \right] \tag{3}$$

$$\leq \frac{1}{n} E\left[ \sum_{i \in I_k} \left\| (\widehat{Z}_{k,i} \widehat{Z}_{k,i}^\top - Z_i Z_i^\top) \right\|^2 \bigg| I_k^c \right] \tag{4}$$

$$= P\left[ \left\| \widehat{Z}_k \widehat{Z}_k^\top - ZZ^\top \right\|^2 \right]. \tag{5}$$

If the RHS of (5) is $o_p(1)$, we can use Lemma 6.1 of [1] to conclude that $\| \frac{1}{\sqrt{n}} \sum_{i \in I_k} W_{k,i} \|$ is $o_p(1)$ as required. Some calculations give

$$\left\| \widehat{Z}_k \widehat{Z}_k^\top - ZZ^\top \right\|^2 \leq 12[(\widehat{g}_k(X) - g_0(X))^2 + (\widehat{g}_k(X)^2 - g_0(X)^2)^2]. \tag{6}$$

Then $P\left[ (\widehat{g}_k - g_0)^2 \right] \leq \sqrt{P[(\widehat{g}_k - g_0)^4]} \to_p 0$. Also

$$P\left[ (\widehat{g}_k^2 - g_0^2)^2 \right] = P[(\widehat{g}_k - g_0)^2 (\widehat{g}_k + g_0)^2] \tag{7}$$

$$\leq \sqrt{P[(\widehat{g}_k - g_0)^4]} \sqrt{P[(\widehat{g}_k + g_0)^4]} \tag{8}$$

$$\leq \sqrt{P[(\widehat{g}_k - g_0)^4]} \sqrt{\sup_{g \in \mathcal{G}} P[g^4]} \tag{9}$$

$$\to_p 0, \tag{10}$$

where the second-to-last line follows because $\widehat{g}_k + g_0 \in \mathcal{G}$ as $\mathcal{G}$ is a vector space. We conclude from (6) that the RHS of (5) is $o_p(1)$.

**Step 2.** By the Cauchy-Schwarz inequality,

$$\left\| \frac{1}{N} \sum_k \sum_{i \in I_k} Z_i(\widehat{g}_k) Y_i \right\| \leq \sqrt{\frac{1}{N} \sum_k \sum_{i \in I_k} \|Z_i(\widehat{g}_k)\|^2} \sqrt{\frac{1}{N} \sum_k \sum_{i \in I_k} Y_i^2}. \tag{11}$$

As $E[Y^2] < \infty$, the second term on the RHS is $O_p(1)$ by Markov's inequality. Also for $i \in I_k$, $E\left[ \|Z_i(\widehat{g}_k)\|^2 \right] = E[1 + T_i + \widehat{g}_k(X_i)^2 + T_i \widehat{g}_k(X_i)^2] \leq \sup_{g \in \mathcal{G}} E[2[1 + g(X_i)^2]] < \infty$, and by Markov's inequality the first term on the RHS is also $O_p(1)$.

**Step 3.** By the central limit theorem, $\sqrt{N}\left[\sum_i \frac{Z_i Z_i^\top}{N} - P[ZZ^\top]\right]$ is asymptotically normal. By the delta method and invertibility of $P[ZZ^\top]$, $\sqrt{N}\left[\left[\sum_i \frac{Z_i Z_i^\top}{N}\right]^{-1} - P[ZZ^\top]^{-1}\right]$ is also, and hence its norm is $O_p(1)$.

**Step 4.** We show that for any $k$, $\frac{1}{n}\sum_{i\in I_k}(\widehat{g}_k(X_i) - g_0(X_i))Y_i = o_p(1)$, from which the result follows. By Cauchy-Schwarz,

$$\frac{1}{n}\sum_{i\in I_k}(\widehat{g}_k(X_i) - g_0(X_i))Y_i \leq \sqrt{\frac{1}{n}\sum_{i\in I_k}(\widehat{g}_k(X_i) - g_0(X_i))^2}\sqrt{\frac{1}{n}\sum_{i\in I_k}Y_i^2}.$$

As $Y$ has finite second moment by assumption, it remains to show the first term on the RHS is $o_p(1)$. We have

$$\frac{1}{n}\sum_{i\in I_k}(\widehat{g}_k(X_i) - g_0(X_i))^2 = \frac{1}{n}\sum_{i\in I_k}\left[(\widehat{g}_k(X_i) - g_0(X_i))^2 - P[(\widehat{g}_k - g_0)^2]\right] + P[(\widehat{g}_k - g_0)^2].$$
(12)

From Lemma 6.1 in [1], the first term on the RHS in (12) is $o_p(1)$ and by the convergence assumption on $\widehat{g}_k$, the second term is too.

**Step 5.** By the continuous mapping theorem it suffices to show that $\|\frac{1}{K}\sum_k\left[P[Z(\widehat{g}_k)Z(\widehat{g}_k)^\top] - P[Z(g_0)Z(g_0)^\top]\right]\| = o_p(1)$. From the argument in Step 1, both $P[(\widehat{g}_k - g_0)^2]$ and $P[(\widehat{g}_k^2 - g_0^2)^2]$ are $o_p(1)$ for all $k$, and hence $P[\widehat{g}_k - g_0]$ and $P[\widehat{g}_k^2 - g_0^2]$ are both $o_p(1)$ for all $k$. The other entries in the matrix are straightforwardly $o_p(1)$.

**Step 6.** This follows from Step 8 and the fact that by Chebyshev's inequality, $\|\frac{1}{\sqrt{N}}\sum_i[Z_iY_i - P[ZY]]\| = O_p(1)$.

**Step 7.** $P[ZZ^\top]$ is invertible by assumption.

**Step 8.** The reasoning here is similar to Step 1. For any $k$ and $i \in I_k$, define $W_{k,i} = \widehat{Z}_{k,i}Y_i - P[\widehat{Z}_kY] - Z_iY_i + P[ZY]$, and note that conditional on the data in $I_k^c$, the $W_{k,i}$ are mean zero matrices, uncorrelated across observations in $I_k$. Then

$$E\left[\left\|\frac{1}{\sqrt{n}}\sum_{i\in I_k}W_{k,i}\right\|^2 \Big| I_k^c\right] \leq \frac{1}{n}E\left[\sum_{i\in I_k}\left\|(\widehat{Z}_{k,i}Y_i - Z_iY_i)\right\|^2 \Big| I_k^c\right] = P\left[\left\|\widehat{Z}_kY - ZY\right\|^2\right].$$

Because $P[(\widehat{g}_k(X) - g_0(X))^2Y^2] \leq \sqrt{P[(\widehat{g}_k - g_0)^4]}\sqrt{P[Y^4]} \to_p 0$, the RHS of (2) is $o_p(1)$. We use Lemma 6.1 of [1] to conclude that $\left\|\frac{1}{\sqrt{n}}\sum_{i\in I_k}W_{k,i}\right\|$ is also $o_p(1)$, from which the result follows.

## 3 Proof of Theorem 1

We have

$$\widehat{\alpha}_1(\{\widehat{g}_k\}_{k=1}^K) - \widehat{\alpha}_1(g_0) = \left[\widehat{\alpha}_1(\{\widehat{g}_k\}_{k=1}^K) - \beta_1(\{\widehat{g}_k\}_{k=1}^K) - \beta_3(\{\widehat{g}_k\}_{k=1}^K)\frac{1}{K}\sum_{k=1}^K P\widehat{g}_k\right]$$
(13)

$$- [\widehat{\alpha}_1(g_0) - \beta_1(g_0) - \beta_3(g_0)Pg_0]$$
(14)

$$= A + B,$$
(15)

where

$$A = [\widehat{\beta_1}(\{\widehat{g}_k\}_{k=1}^K) - \beta_1(\{\widehat{g}_k\}_{k=1}^K)] - [\widehat{\beta_1}(g_0) - \beta_1(g_0)],$$
(16)

and

$$B = \underbrace{\left[\widehat{\beta_3}(\{\widehat{g}_k\}_{k=1}^K)\frac{1}{N}\sum_i \widehat{g}_{k(i)}(X_i) - \beta_3(\{\widehat{g}_k\}_{k=1}^K)\frac{1}{K}\sum_{k=1}^K P\widehat{g}_k\right]}_{C}$$
$$- \underbrace{\left[\widehat{\beta_3}(g_0)\frac{1}{N}\sum_i g_0(X_i) - \beta_3(g_0)Pg_0\right]}_{D}. \tag{17}$$

Proposition 1 has established that $A = o_p(1/\sqrt{N})$. Moreover

$$C = \underbrace{\left(\widehat{\beta_3}(\{\widehat{g}_k\}_{k=1}^K) - \beta_3(\{\widehat{g}_k\}_{k=1}^K)\right)\frac{1}{N}\sum_i \widehat{g}_{k(i)}(X_i)}_{C_1} + \underbrace{\beta_3(\{\widehat{g}_k\}_{k=1}^K)\left(\frac{1}{N}\sum_i \left[\widehat{g}_{k(i)}(X_i) - P\widehat{g}_{k(i)}\right]\right)}_{C_2} \tag{18}$$

and

$$D = \underbrace{\left(\widehat{\beta_3}(g_0) - \beta_3(g_0)\right)\frac{1}{N}\sum_i g_0(X_i)}_{D_1} + \underbrace{\left(\beta_3(g_0)\frac{1}{N}\sum_i [g_0(X_i) - Pg_0]\right)}_{D_2}. \tag{19}$$

We show $C_1 - D_1$ and $C_2 - D_2$ are $o_p(1/\sqrt{N})$ to conclude. In fact

$$C_1 - D_1 = \left(\widehat{\beta_3}(\{\widehat{g}_k\}_{k=1}^K) - \beta_3(\{\widehat{g}_k\}_{k=1}^K) - \widehat{\beta_3}(g_0) + \beta_3(g_0)\right)\frac{1}{N}\sum_i \widehat{g}_{k(i)}(X_i)$$
$$+ \left(\widehat{\beta_3}(g_0) - \beta_3(g_0)\right)\frac{1}{N}\sum_i \left[\widehat{g}_{k(i)}(X_i) - g_0(X_i)\right] = o_p(1/\sqrt{N}). \tag{20}$$

This is because

- $\widehat{\beta_3}(\{\widehat{g}_k\}_{k=1}^K) - \beta_3(\{\widehat{g}_k\}_{k=1}^K) - \widehat{\beta_3}(g_0) + \beta_3(g_0) = o_p(1/\sqrt{N})$ from Proposition 1;
- $\frac{1}{N}\sum_i \widehat{g}_{k(i)}(X_i) = \frac{1}{N}\sum_i g_0(X_i) + \frac{1}{N}\sum_i(\widehat{g}_{k(i)}(X_i)) - g_0(X_i)) = O_p(1)$ from the LLN and the same logic bounding (12) above;
- $\widehat{\beta_3}(g_0) - \beta_3(g_0) = O_p(1/\sqrt{N})$ from the CLT and the fact that $P(Z(g_0)Z(g_0)\top)$ has all eigenvalues bounded away from 0;
- $\frac{1}{N}\sum_i(\widehat{g}_{k(i)}(X_i) - g_0(X_i)) = o_p(1)$ again from bounding argument applied to (12).

Similarly,

$$C_2 - D_2 = \beta_3(\{\widehat{g}_k\}_{k=1}^K)\left(\frac{1}{N}\sum_i \left[\left[\widehat{g}_{k(i)}(X_i) - P\widehat{g}_{k(i)}\right] - [g_0(X_i) - Pg_0]\right]\right)$$
$$+ \left((\beta_3(\{\widehat{g}_k\}_{k=1}^K) - \beta_3(g_0))\frac{1}{N}\sum_i [g_0(X_i) - Pg_0]\right) = o_p(1/\sqrt{N}), \tag{21}$$

which results from the following facts:

- $\beta_3(\{\widehat{g}_k\}_{k=1}^K) = \beta_3(g_0) + (\beta_3(\{\widehat{g}_k\}_{k=1}^K) - \beta_3(g_0)) = O_p(1)$;
- $\frac{1}{N}\sum_i \left[\left[\widehat{g}_{k(i)}(X_i) - P\widehat{g}_{k(i)}\right] - [g_0(X_i) - Pg_0]\right] = o_p(1/\sqrt{N})$ from the same reasoning applied to bound (1);
- $\beta_3(\{\widehat{g}_k\}_{k=1}^K) - \beta_3(g_0) = o_p(1)$ due to convergence of $\widehat{g}_k$ to $g_0$, continuity of $\beta_3(\cdot)$, and the continuous mapping theorem;
- $\frac{1}{N}\sum_i [g_0(X_i) - Pg_0] = O_p(1/\sqrt{N})$ from the CLT.

Combining the above arguments, we conclude that $B = o_p(1/\sqrt{N})$.

# 4 Proof of Proposition 4

We first show that $\widehat{Var}(\widehat{g}_{k(i)}(X_i)) \to_p \sigma_g^2$. We have

$$\widehat{Var}(\widehat{g}_{k(i)}(X_i)) = \frac{1}{K}\sum_k \frac{1}{n}\sum_{i\in I_k}\widehat{g}_k(X_i)^2 - \left[\frac{1}{K}\sum_k \frac{1}{n}\sum_{i\in I_k}\widehat{g}_k(X_i)\right]^2. \tag{22}$$

By the same logic as in Step 1 of the proof of Proposition 1, for each $k = 1, 2, \ldots, K$,

$$E\left[\left\|\frac{1}{n}\sum_{i\in I_k}[\widehat{g}_k(X_i)^2 - P\widehat{g}_k^2]\right\|^2 \Big| I_k^c\right] \to_p 0,$$

and so $\frac{1}{n}\sum_{i\in I_k}\widehat{g}_k(X_i)^2 - P\widehat{g}_k^2 \to_p 0$. Since $P\widehat{g}_k^2 \to_p Pg_0^2$, it follows that $\frac{1}{n}\sum_{i\in I_k}\widehat{g}_k(X_i)^2 \to_p Pg_0^2$. Similarly $\frac{1}{n}\sum_{i\in I_k}\widehat{g}_k(X_i) \to_p Pg_0$. Hence $\widehat{Var}(\widehat{g}_{k(i)}(X_i)) \to_p \sigma_g^2$. Also, by Proposition 1,

$$\left\|\widehat{\beta}(\{\widehat{g}_k\}_{k=1}^K) - \beta(\{\widehat{g}_k\}_{k=1}^K)\right\| \to_p 0 \tag{23}$$

and by continuity of $\beta(\cdot)$ and the continuous mapping theorem,

$$\left\|\beta(\{\widehat{g}_k\}_{k=1}^K) - \beta(g_0)\right\| \to_p 0. \tag{24}$$

Consequently $\left\|\widehat{\beta}(\{\widehat{g}_k\}_{k=1}^K) - \beta(g_0)\right\| \to_p 0$. By the continuous mapping theorem, we conclude that $\widehat{\sigma}^2 \to_p \sigma^2$.

# 5 Proof of auxiliary lemmas

**Lemma 1.** *Given Assumption 1,*

$$\left\|\frac{1}{N}\sum_k \sum_{j\in I_k}\widehat{Z}_{k,j}\widehat{Z}_{k,j}^\top - \frac{1}{K}\sum_k P(\widehat{Z}_k\widehat{Z}_k^\top)\right\| = O_p(1/\sqrt{n}).$$

*Proof.* Since the number of splits $K$ is bounded, we only need to verify for any $k \in \{1, 2, \ldots, K\}$,

$$\left\|\frac{1}{n}\sum_{j\in I_k}\widehat{Z}_{k,j}\widehat{Z}_{k,j}^\top - P(\widehat{Z}_k\widehat{Z}_k^\top)\right\| = O_p(1/\sqrt{n}).$$

Below we'll prove

$$\frac{1}{n}\sum_{j\in I_k}T_j^2\widehat{g}_k^2(X_j) - E[T_j^2\widehat{g}_k^2(X_j)|I_k^c] = O_p(1/\sqrt{n}). \tag{25}$$

The other terms can be derived in the similar manner.

First, since $P(\widehat{g}_k - g_0)^4 \to_p 0$ as $n \to \infty$, we know that for any subsequence $\{n_l\}$ of $\mathbb{N}$, it further has a subsequence $\{n_l'\}$, such that $P(\widehat{g}_k - g_0)^4 \to 0$ a.s. as $l \to \infty$. Our next step is to prove

$$\frac{1}{\sqrt{n_l'}}\sum_{j\in I_k}T_j^2\widehat{g}_k^2(X_j) - E[T_j^2\widehat{g}_k^2(X_j)|I_k^c] = O_p(1) \tag{26}$$

as $l \to \infty$.

For notational simplicity, define $V_{k,j} := T_j^2\widehat{g}_k^2(X_j) - E[T_j^2\widehat{g}_k^2(X_j)|I_k^c]$. Since $\{V_{k,j}\}_{j\in I_k}$ are independent conditioned on $I_k^c$, for any $t \in \mathbb{R}$ we have

$$E\exp\left(it/\sqrt{n_l'}\cdot\sum_{j\in I_k}V_{k,j}\right) = EE\left[\exp\left(it/\sqrt{n_l'}\cdot\sum_{j\in I_k}V_{k,j}\right)\Big|I_k^c\right]$$

$$= E\left\{E\left[\exp\left(it/\sqrt{n_l'}\cdot V_{k,j}\right)\Big|I_k^c\right]\right\}^{n_l'}.$$

Furthermore,

$$\lim_{l \to \infty} E \exp\left(it/\sqrt{n_l'} \cdot \sum_{j \in I_k} V_{k,j}\right) = \lim_{l \to \infty} E\left\{E\left[\exp\left(it/\sqrt{n_l'} \cdot V_{k,j}\right)\Big|I_k^c\right]\right\}^{n_l'}$$

$$= E \lim_{l \to \infty} \left\{E\left[\exp\left(it/\sqrt{n_l'} \cdot V_{k,j}\right)\Big|I_k^c\right]\right\}^{n_l'}. \tag{27}$$

Our goal is now to derive the limit in the last term so that we can infer the limiting distribution of $1/\sqrt{n_l'} \cdot \sum_{j \in I_k} V_{k,j}$.

First, we conduct the Taylor expansion

$$\exp\left(it/\sqrt{n_l'} \cdot V_{k,j}\right) = 1 + it.\sqrt{n_l'} \cdot V_{k,j} - \frac{t^2}{2n_l'}V_{k,j}^2 + R_{k,j}.$$

Here

$$R_{k,j} = \exp\left(it/\sqrt{n_l'} \cdot V_{k,j}\right) - \left[1 + it/\sqrt{n_l'} \cdot V_{k,j} - \frac{t^2}{2n_l'}V_{k,j}^2\right].$$

Thus

$$E\left[\exp\left(it/\sqrt{n_l'} \cdot V_{k,j}\right)\Big|I_k^c\right] = 1 + it/\sqrt{n_l'} \cdot E[V_{k,j}|I_k^c] -$$

$$\frac{t^2}{2n_l'}E[V_{k,j}^2|I_k^c] + E[R_{k,j}|I_k^c] = 1 - \frac{t^2}{2n_l'}E[V_{k,j}^2|I_k^c] + E[R_{k,j}|I_k^c] \tag{28}$$

First, with probability 1,

$$\lim_{l \to \infty} E[V_{k,j}^2|I_k^c] = \lim_{l \to \infty} \left\{E[T_j^4\widehat{g}_k^4(X_j)|I_k^c] - E[T_j^2\widehat{g}_k^2(X_j)|I_k^c]^2\right\}$$

$$= p \cdot Pg_0^4 - p^2 \cdot (Pg_0^2)^2. \tag{29}$$

Next, we bound $|E[R_{k,j}|I_k^c]|$. In fact,

$$R_{k,j} \leq \begin{cases} \frac{2t^3}{n_l'^{3/2}}V_{k,j}^3 & \text{when } |V_{k,j}| \leq \frac{\sqrt{n_l'}}{2t}, \\ 2 + \frac{t}{\sqrt{n_l'}}|V_{k,j}| + \frac{t^2}{2n_l'}|V_{k,j}|^2 & \text{otherwise.} \end{cases}$$

This means

$$|E[R_{k,j}|I_k^c]| \leq E[R_{k,j}^{(1)}|I_k^c] + E[R_{k,j}^{(2)}|I_k^c],$$

where $R_{k,j}^{(1)} = \frac{2t^3}{n_l'^{3/2}}|V_{k,j}|^3 1_{\{|V_{k,j}| \leq \sqrt{n_l'}/(2t)\}}$,
$R_{k,j}^{(2)} = (2 + \frac{t}{\sqrt{n_l'}}|V_{k,j}| + \frac{t^2}{2n_l'}|V_{k,j}|^2)1_{\{|V_{k,j}| > \sqrt{n_l'}/(2t)\}}$.

On the one hand,

$$E[R_{k,j}^{(1)}|I_k^c] \leq \frac{2t^3}{n_l'^{3/2}}E\left[|V_{k,j}|^{2+\delta/2} \cdot \left(\sqrt{n_l'}/2t\right)^{1-\delta/2}\Big|I_k^c\right]$$

$$= \frac{2^{\delta/2}t^{2+\delta/2}}{n_l'^{1+\delta/4}}E\left[|T_j^2\widehat{g}_k^2(X_j) - ET_j^2\widehat{g}_k^2(X_j)|^{2+\delta/2}\Big|I_k^c\right] \leq \frac{2^{2+\delta}t^{2+\delta/2}}{n_l'^{1+\delta/4}}P|\widehat{g}_k|^{4+\delta}.$$

On the other hand, by Markov's inequality,

$$E[R_{k,j}^{(2)}|I_k^c] \leq 2E\left[\left(2t/\sqrt{n_l'}\right)^{2+\delta/2}|V_{k,j}|^{2+\delta/2}\Big|I_k^c\right] + t/\sqrt{n_l'} \cdot$$

$$E\left[|V_{k,j}| \cdot \left(2t/\sqrt{n_l'}\right)^{1+\delta/2}|V_{k,j}|^{1+\delta/2}\Big|I_k^c\right] + \frac{t^2}{2n_l'} \cdot$$

$$E\left[|V_{k,j}|^2 \cdot \left(2t/\sqrt{n_l'}\right)^{\delta/2}|V_{k,j}|^{\delta/2}\Big|I_k^c\right] \leq \frac{2^{6+\delta}t^{2+\delta/2}}{n_l'^{1+\delta/4}}P|\widehat{g}_k|^{4+\delta}.$$

Combining the above two bounds, we deduce that

$$|E[R_{k,j}|I_k^c]| \le \frac{2^{7+\delta}t^{2+\delta/2}}{n_l'^{1+\delta/4}} P|\widehat{g}_k|^{4+\delta}.$$

Thus with probability 1, $E[R_{k,j}|I_k^c] = o(1/n_l')$.

Combining the above bound, (28) and (29), we obtain that with probability 1,

$$\lim_{l \to \infty} n_l' \log E\left[ \exp\left( it/\sqrt{n_l'} \cdot V_{k,j} \right) \Big| I_k^c \right]$$
$$= \lim_{l \to \infty} n_l' \log \left( 1 - \frac{t^2}{2n_l'} E[V_{k,j}^2|I_k^c] + E[R_{k,j}|I_k^c] \right)$$
$$= -\frac{t^2}{2n_l'}[p \cdot Pg_0^4 - p^2 \cdot (Pg_0^2)^2].$$

Finally we plug the above into (27) and conclude that

$$\lim_{l \to \infty} E \exp\left( it/\sqrt{n_l'} \cdot \sum_{j \in I_k} V_{k,j} \right) = \exp\left\{ -\frac{t^2}{2n_l'}[p \cdot Pg_0^4 - p^2 \cdot (Pg_0^2)^2] \right\}.$$

This implies that $\frac{1}{\sqrt{n_l'}} \sum_{j \in I_k} V_{k,j}$ converges in distribution to a centered normal random variable with variance $p \cdot Pg_0^4 - p^2 \cdot (Pg_0^2)^2$, and (26) follows.

Finally, since for any subsequence $\{n_l\}$ of $\mathbb{N}$, it further has a subsequence $\{n_l'\}$ such that (26) holds, it can only be the case that (25) is true.

$\square$

**Lemma 2.** *The following hold with probability tending to 1:*

$$\lambda_{\min}\left( \frac{1}{n} \sum_{i \in I_k} \widehat{Z}_{k,i}\widehat{Z}_{k,i}^\top \right) \ge \frac{1}{2} \inf_{g \in \mathcal{G}} \lambda_{min}(P[Z(g)Z(g)^\top]) \quad \forall k \in \{1, 2, \ldots, K\}; \qquad (30)$$

$$\lambda_{\min}\left( \frac{1}{N} \sum_{i=1}^{N} \widehat{Z}_i\widehat{Z}_i^\top \right) \ge \frac{1}{2} \inf_{g \in \mathcal{G}} \lambda_{min}(P[Z(g)Z(g)^\top]). \qquad (31)$$

*Proof.* According to Weyl's inequality,

$$\lambda_{\min}\left( \frac{1}{n} \sum_{i \in I_k} \widehat{Z}_{k,i}\widehat{Z}_{k,i}^\top \right) \ge \lambda_{\min}(P(\widehat{Z}_k\widehat{Z}_k^\top)) - \left\| \frac{1}{n} \sum_{j \in I_k} \widehat{Z}_{k,j}\widehat{Z}_{k,j}^\top - P(\widehat{Z}_k\widehat{Z}_k^\top) \right\|$$
$$\ge \inf_{g \in \mathcal{G}} \lambda_{min}(P[Z(g)Z(g)^\top]) - \left\| \frac{1}{n} \sum_{j \in I_k} \widehat{Z}_{k,j}\widehat{Z}_{k,j}^\top - P(\widehat{Z}_k\widehat{Z}_k^\top) \right\|.$$

On the other hand, from the proof of Lemma 1 we know

$$\left\| \frac{1}{n} \sum_{j \in I_k} \widehat{Z}_{k,j}\widehat{Z}_{k,j}^\top - P(\widehat{Z}_k\widehat{Z}_k^\top) \right\| = O_p(1/\sqrt{n}).$$

This implies that

$$\lim_{n \to \infty} P\left( \left\| \frac{1}{n} \sum_{j \in I_k} \widehat{Z}_{k,j}\widehat{Z}_{k,j}^\top - P(\widehat{Z}_k\widehat{Z}_k^\top) \right\| \ge \frac{1}{2} \inf_{g \in \mathcal{G}} \lambda_{min}(P[Z(g)Z(g)^\top]) \right) = 0.$$

Combining the above, we obtain (30). (31) can be proved in a similar way. $\square$

**Lemma 3.** *Let* $\{M_{1n}\}, \{M_{2n}\}, \{M_{3n}\}, \{M_{4n}\}, \{A_n\}, \{B_n\}$ *be sequences of random real symmetric matrices of fixed dimension. Assume that with probability 1,* $\lambda_0 := \inf_n \lambda_{\min}(B_n) > 0$, *and* $\|A_n - B_n\| = o_p(1)$. *Moreover, assume that*

$$\|M_{1n} - A_n\| = O_p(1/\sqrt{n}), \|M_{3n} - A_n\| = O_p(1/\sqrt{n}),$$
$$\|M_{2n} - B_n\| = O_p(1/\sqrt{n}), \|M_{4n} - B_n\| = O_p(1/\sqrt{n}).$$

*If in addition,*

$$\sqrt{n}\|M_{1n} + M_{2n} - M_{3n} - M_{4n}\| \to_p 0,$$

*then*

$$\sqrt{n}\|M_{1n}^{-1} + M_{2n}^{-1} - M_{3n}^{-1} - M_{4n}^{-1}\| \to_p 0.$$

*Proof.* Define the event

$$E_n := \{\|A_n - B_n\| \geq \lambda_0/2\} \cup \{\max\{\|M_{1n} - A_n\|, \|M_{3n} - A_n\|\} \geq \lambda_0/2\}$$
$$\cup \{\max\{\|M_{2n} - B_n\|, \|M_{4n} - B_n\|\} \geq \lambda_0/2\}.$$

Then $\lim_{n\to\infty} P(E_n) = 0$. Now on $E_n^c$, according to a Neumann series expansion,

$$M_{1n}^{-1} = [A_n + (M_{1n} - A_n)]^{-1}$$
$$= A_n^{-1/2}[I - A_n^{-1/2}(M_{1n} - A_n)A_n^{-1/2} + D_{1n}]A_n^{-1/2}.$$

Here $D_{1n} = \sum_{j\geq 2}[-A_n^{-1/2}(M_{1n} - A_n)A_n^{-1/2}]^j$, and we have on $E_n^c$

$$\|D_{1n}\| \leq \sum_{j\geq 2}\|A_n^{-1/2}(M_{1n} - A_n)A_n^{-1/2}\|^j$$
$$\leq \frac{\|A_n^{-1}\|^2\|M_{1n} - A_n\|^2}{1 - \|A_n^{-1}\|\|M_{1n} - A_n\|} \leq \frac{8}{\lambda_0^2}\|M_{1n} - A_n\|^2. \tag{32}$$

Here we use the fact that on $E_n^c$

$$\|A_n^{-1/2}(M_{1n} - A_n)A_n^{-1/2}\| \leq \|A_n^{-1/2}\|^2\|M_{1n} - A_n\| < \frac{2}{\lambda_0} \cdot \frac{\lambda_0}{2} = 1.$$

Similar expansions hold for $M_{2n}$, $M_{3n}$ and $M_{4n}$, and we define $D_{2n}$, $D_{3n}$ and $D_{4n}$ accordingly. Using some simple algebra, we deduce that on $E_n^c$,

$$M_{1n}^{-1} + M_{2n}^{-1} - M_{3n}^{-1} - M_{4n}^{-1} = J_{1n} + J_{2n} + J_{3n} + J_{4n},$$

where

$$J_{1n} = -A_n^{-1}[M_{1n} + M_{2n} - M_{3n} - M_{4n}]A_n^{-1},$$
$$J_{2n} = -A_n^{-1}(M_{4n} - M_{2n})A_n^{-1} + B_n^{-1}(M_{4n} - M_{2n})B_n^{-1},$$
$$J_{3n} = A_n^{-1/2}(D_{1n} - D_{3n})A_n^{-1/2},$$
$$J_{4n} = B_n^{-1/2}(D_{2n} - D_{4n})B_n^{-1/2}.$$

For any $\epsilon > 0$,

$$P(\sqrt{n}\|M_{1n}^{-1} + M_{2n}^{-1} - M_{3n}^{-1} - M_{4n}^{-1}\| > \epsilon) < P(E_n) + \sum_{\ell=1}^{4} P(E_n^c \cap \{\sqrt{n}\|J_{\ell n}\| > \epsilon/4\}). \tag{33}$$

Combining the fact that $\lim_{n\to\infty} P(E_n) = 0$, we only need to prove that each of the rest of the terms on the the RHS of (33) has limit 0.

First, $\lim_{n\to\infty} P(E_n^c \cap \{\sqrt{n}\|J_{1n}\| > \epsilon/4\}) = 0$ follows from our assumption. For $J_{2n}$, observe that $J_{2n} = J_{2n}^{(1)} + J_{2n}^{(2)}$, where

$$J_{2n}^{(1)} = (B_n^{-1} - A_n^{-1})(M_{4n} - M_{2n})A_n^{-1}, J_{2n}^{(2)} = B_n^{-1}(M_{4n} - M_{2n})(B_n^{-1} - A_n^{-1}).$$

We bound the limit of $\|J_{2n}^{(1)}\|$ as follows: For any $\delta > 0$, there exists $M > 0$ such that $\forall n$, $P(\sqrt{n}\|M_{4n} - M_{2n}\| > M) < \frac{\delta}{2}$. According to our assumption, there further exists $N \in \mathbb{N}$ such that for all $n > N$, $P(\|A_n - B_n\| > \frac{\lambda_0^3 \epsilon}{32M}) < \frac{\delta}{2}$. Therefore for all $n > N$,

$$P(E_n^c \cap \{\sqrt{n}\|J_{2n}^{(1)}\| > \epsilon/8\})$$
$$\leq P(E_n^c \cap \{\sqrt{n}\|A_n^{-1}(A_n - B_n)B_n^{-1}(M_{4n} - M_{2n})A_n^{-1}\| > \epsilon/8\})$$
$$\leq P(E_n^c \cap \{\|A_n - B_n\| \cdot \sqrt{n}\|M_{4n} - M_{2n}\| > \lambda_0^3 \epsilon/32\})$$
$$\leq P(\sqrt{n}\|M_{4n} - M_{2n}\| > M) + P(\|A_n - B_n\| > \lambda_0^3 \epsilon/(32M)) < \delta.$$

The above argument implies that $\lim_{n \to +\infty} P(E_n^c \cap \{\sqrt{n}\|J_{2n}^{(1)}\| > \epsilon/8\}) = 0$. Similarly we have $\lim_{n \to +\infty} P(E_n^c \cap \{\sqrt{n}\|J_{2n}^{(2)}\| > \epsilon/8\}) = 0$. Thus

$$\lim_{n \to +\infty} P(E_n^c \cap \{\sqrt{n}\|J_{2n}\| > \epsilon/4\})$$
$$\leq \lim_{n \to +\infty} P(E_n^c \cap \{\sqrt{n}\|J_{2n}^{(1)}\| > \epsilon/8\}) + \lim_{n \to +\infty} P(E_n^c \cap \{\sqrt{n}\|J_{2n}^{(2)}\| > \epsilon/8\}) = 0.$$

Now we proceed to bound the limit of $\|J_{3n}\|$. In fact we have

$$P(E_n^c \cap \{\sqrt{n}\|J_{3n}\| > \epsilon/4\}) \leq P(E_n^c \cap \{\sqrt{n}\|D_{1n} - D_{3n}\| > \epsilon\lambda_0/8\})$$
$$\leq P(E_n^c \cap \{\sqrt{n}\|D_{1n}\| > \epsilon\lambda_0/16\}) + P(E_n^c \cap \{\sqrt{n}\|D_{3n}\| > \epsilon\lambda_0/16\})$$
$$\leq P(\sqrt{n}\|M_{1n} - A_n\|^2 > \epsilon\lambda_0^3/128) + P(\sqrt{n}\|M_{3n} - A_n\|^2 > \epsilon\lambda_0^3/128).$$

In the last inequality we utilize (32). Combining our assumptions, we have

$$\lim_{n \to \infty} P(E_n^c \cap \{\sqrt{n}\|J_{3n}\| > \epsilon/4\}) = 0.$$

Similarly

$$\lim_{n \to \infty} P(E_n^c \cap \{\sqrt{n}\|J_{4n}\| > \epsilon/4\}) = 0.$$

We conclude our proof. $\qquad \square$

## References

[1] CHERNOZHUKOV, Victor ; CHETVERIKOV, Denis ; DEMIRER, Mert ; DUFLO, Esther ; HANSEN, Christian ; NEWEY, Whitney ; ROBINS, James: Double/debiased machine learning for treatment and structural parameters. In: *The Econometrics Journal* 21 (2018), Nr. 1