# OpenReview forum: "Machine Learning for Variance Reduction in Online Experiments"
_NeurIPS.cc/2021/Conference — NeurIPS 2021 Poster_

### Official Review · Reviewer_SmDS · 2021-06-25

**Rating:** 7
**Confidence:** 3

**Summary:**

This paper proposes a new approach for variance reduction in randomized controlled trials. The proposed method first predicts the outcome variable Y from a set of covariates X that are independent of the treatment assignment T, and measures the average treatment effect by regressing the difference between this prediction and the observed Y against the predicted $\hat{Y}$. The method avoids the potential pitfall of invalid confidence intervals by training the predictor for a given $Y_i$ using data from $j \neq i$ ("cross-fitting"), and avoids the potential pitfall of over-reliance on poor estimates of Y by including the predictions as part of a linear regression (instead of directly subtracting the predictions from the observed Y). The consistency and asymptotic normality of the estimator are proven, and experiments on both simulated and real data are presented.

**Limitations And Societal Impact:**

Yes

**Main Review:**

This paper does a good job of placing itself within the existing literature and improving one part of existing methods. Specifically, existing work [18] already presented an OLS estimate of the average treatment effect; this work adds additional flexibility to the terms of the OLS estimate, to allow for arbitrary predictors of $Y_i$ using the features $X_i$. The paper is well-scoped, discussing both the theoretical aspects of the estimator (asymptotic properties, correct confidence intervals) and demonstrating success empirically. I particularly liked how the exposition and notation clearly emphasized the important parts of the asymptotic analysis: specifically, the convergence of functions of $\hat{g}_k$ to functions of $g_0$.

The only aspect of the estimator that I would have liked to see discussed in more detail was the choice of the cross-fitting hyperparameter $K$. The authors note in Remark 4 that the choice of $K$ may matter in finite samples although it isn't important asymptotically. I had imagined that the choice $K=N$ would have given the largest varince reduction (since the most data is then used to train each $\hat{g}_k$), so I was surprised to see that $K=2$ was the only choice considered for the experiments. I would appreciate some discussion of the tradeoffs in choosing $K$ in finite samples - are there any downsides to increasing $K$ besides increased computation?

I also have several smaller comments for the authors:
- In both the simulated and real data results, the proposed method is compared to the differences-in-means baseline and the CUPED baseline. My reading of the experiments section is that the baselines got to use all $N$ data points (ie, they were not subject to cross-fitting), whereas the MLRATE estimator trained each $\hat{g}_k$ on $N/2$ data points. I just wanted to confirm that this is how the experiment was run, since it has implications for the interpretation of the later claim that "the univariate procedure would require sample sizes 1.56 times as large."
- On line 188, the word "the" is repeated
- Figure 1a really only shows three numbers (the CI can be derived from the variance and vice versa). I wonder if there is a more meaningful way to show the data in Figure 1. For example, you could extend Figure 1b to "facet" in two directions, "estimator" ~ "baseline", and then there would be a blank plot in "CUPED" x "CUPED". This is just one suggestion, and by no means the only one, but I would encourage you to think about how to best use the space you have available to communicate your results.
- I found the paper very readable, thank you for spending time on the writing.

# Response to author response

Thank you for your thoughtful response, especially regarding the choice of $K$. I appreciated your detailed consideration of this question, and remain supportive of the work.

**Time Spent Reviewing:**

2

---

> ### Author Response · Authors · 2021-08-10
> **Response to Review Four**
>
> We thank the reviewer for their excellent feedback and suggestions, which will help us improve the paper, and we very much appreciate their careful reading and evaluation of this work.
>
> **Choice of $K$.**
>
> We fully agree that the discussion of the choice of $K$ in the paper was regrettably brief, and there was no data-driven evaluation of the finite-sample properties of different choices of $K$. As mentioned in the review, computation is a consideration, and the computational cost will often be superlinear in $K$. It’s certainly possible that larger values of K may produce better results in finite samples, and our intention was to show that even for the least expensive choice of $K = 2$, our method performs well. That is, estimating many ML models is not a necessary condition for successful application of this methodology.
>
> As Chernozhukov et al [2018] note, if the prediction problem is complex, it may benefit from putting more data in the training splits, and they suggest $K = 4$ or 5. Much larger values of $K$ may be unattractive given diminishing returns in model performance and the extra compute cost. Choosing a $K$ which grows with the sample size, such as $K = N$, is not covered by our current theoretical analysis, which assumes a fixed $K$.
>
> We conducted the simulation with the same data generating process as in our paper, for $K \in \\{2, 5, 10, 20\\}$, $N = 1000$, with 50 covariates and for 1000 simulation repetitions per value of $K$. $95\\%$ confidence intervals are computed for the same procedures considered in our paper. The following two tables show the Average CI Coverage as well as relative CI width compared to that of the difference-in-means estimators. We observe that all procedures provide confidence intervals that cover the true ATE about $95\\%$ of the time with any of the $K$ values. For the two MLRATE procedures, larger $K$ leads to slightly shorter confidence intervals, and the gain tends to diminish after $K > 10$.
>
>     Table 1: Average CI Coverage
>                          K=2       K=5      K=10      K=20
>     MLRATE_GBDT          0.941     0.943    0.941     0.948
>     MLRATE-Elastic Net   0.936     0.940    0.940     0.947
>     Unadjusted           ----------     0.938    ----------
>
>     Table 2: Average Relative CI Width
>                          K=2       K=5      K=10      K=20
>     MLRATE_GBDT          0.783     0.771    0.768     0.766
>     MLRATE-Elastic Net   0.871     0.867    0.865     0.865
>     Unadjusted           ----------     1.000    ----------
>
> Finally, we note that as is generally the case in applications of K-fold cross validation for supervised learning problems, it may be beneficial to average over different random partitions of the data into K-folds. Heuristically this can be motivated as a form of Rao-Blackwellization, in which it cannot hurt performance to average out random variation unrelated to the data generating process of interest.
>
> **Remaining Comments.**
>
> In response to the remaining comments:
> - That is correct. The baseline methods did not use cross-fitting, and were applied to the full dataset of $N$ observations. The MLRATE procedures used the same dataset of $N$ observations, and each trained two models on subdatasets of size $N/2$.
> - Thank you for pointing this out, we will correct this typo.
> - Thank you for the comment. Indeed, we could alternatively present a facet plot showing pairwise comparisons of estimator performance, as measured in terms of relative variance or confidence interval width. To make this plot more information-dense we could also add more information about the distribution of performance across the 48 metrics we study, for example with indicators for the 25th, 50th and 75th percentiles of the distribution of relative variance across metrics in Figure 1a.
> - Thank you for the kind feedback. Our hope is that other practitioners will adopt this methodology and find it useful, as has been the case at Facebook. To that end we tried to make the exposition as clear as possible.

---

### Official Review · Reviewer_dXBW · 2021-07-16

**Rating:** 9
**Confidence:** 5

**Summary:**

This paper considers the analysis of randomized (online) experiments (A/B tests) through a linear regression that adjusts for pre-treatment covariates, which are independent of the treatment assignment. It is well-known that this type of estimate of the treatment effect can be more efficient than the difference-in-means estimator when the covariates are predictive of the outcome.

The authors propose to incorporate the covariates into the linear regression through a predictor of the outcome, which itself is estimated with data-adaptive, machine learning methods. Cross-fitting is used to control the bias introduced by the data-adaptive estimation. It is shown that the resulting estimator is asymptotically normal with a limiting variance that is no larger than that of the difference-in-mean estimator, under relatively weak conditions. In particular, the method does not require the consistency of the nuisance predictor function. This model-agnosticism is clearly an advantage when compared to influence-function-based methods (e.g., double ML), which typically require the convergence of the nuisance to the truth at a certain rate.

The method developed is shown to deliver correct coverage from simulations and A/A testing data, and to typically outperform competing methods.

**Limitations And Societal Impact:**

I do not see any potential negative societal impact of the work in its current form.

**Main Review:**

I think this is an excellent paper that delivers an easy-to-use method for the analysis of A/B tests with strong inferential guarantees. The method is able to incorporate machine learning methods flexibly to yield tighter confidence intervals, which can be higly valuable given the typical small effect sizes in well-optimized industrial settings.

### Main comments

1. **Semiparametric efficiency**: It might be interesting to compare the method with one that estimates the effect based on the semiparametric efficient influence function. I like the comparison made for the simulation data but would like to understand the more general case.

It can be argued that the downside of the model-agnosticism is that the method might not achieve the semiparametric efficiency bound. I am wondering, when the nuisance happens to be consistent (at a certain rate), can anything be said relative to the efficiency bound?

2. I went through the technical proofs (though not exhaustively) --- the results are clearly stated and seem correct to me.

### Minor comments

1. Is there any guidance on choosing the number of splits for cross-fitting?

**Time Spent Reviewing:**

2

---

> ### Author Response · Authors · 2021-08-10
> **Response to Review Three**
>
> We thank this reviewer for their feedback and reading of our manuscript, and we are thrilled that the reviewer thinks highly of the work!
>
> We fully agree that the price to pay for model agnosticism is sacrificing the kind of efficiency guarantees one might obtain with a well-specified model. We offer the following remarks regarding the estimator based on the semiparametric efficient influence function, and its comparison to our proposed estimator.
> - For the estimator based on the semiparametric efficient influence function, if we do have consistency of the nuisance parameters (which in this case corresponds to the two conditional mean functions, one for the test and one for the control group), then this estimator is not only consistent and asymptotically normal, it is also efficient. One may obtain this result as a special case of the double ML framework studied in Chernozhukov et al [2018] (see especially the discussion on pg 25 and 26 of that paper).
> - Interestingly the above result is true *regardless* of the rate at which the nuisance parameters are consistently estimated. This is essentially because the propensity score is known here, which allows us to do better than the $n^{-¼}$ rate requirement in the more general case. An alternative, equally valid interpretation is that this improvement is possible because the scores satisfy a stronger condition than Neyman-orthogonality: not only is the expected (uncentered) efficient influence function locally insensitive to the value of the nuisance parameter to first-order, it in fact doesn’t depend on the nuisance parameter *at all*.
> - The case of a nuisance parameter estimator that converges but is inconsistent is not treated in the double ML paper. We might expect that the estimator based on the efficient influence function is still consistent and asymptotically normal, but can no longer hope for semiparametric efficiency. Without further assumptions this estimator may perform arbitrarily poorly. With poorly-specified conditional mean estimators, the “adjustment” may do more harm than good, and add extra variance relative to the difference-in-means estimator. Indeed this can be seen as a motivation for our proposed estimator: the linear regression step we add is a safeguard against poorly-specified conditional mean estimators, and guarantees non-inferiority relative to the difference-in-means estimator despite such misspecification.
>
> Thank you for raising the comment regarding the choice of the number of splits for cross-fitting. A related question was also raised in Review 4. We comment in more detail on this choice there, and also provide quantitative evidence in the form of estimator performance in simulations with varying choices of $K$.

---

### Official Review · Reviewer_CjHF · 2021-07-16

**Rating:** 5
**Confidence:** 4

**Summary:**

The setting of this paper is that of a randomized controlled trial with data $(Y_i, X_i, T_i)$ for $i=1,...,N$. $Y_i$ is the response of interest, $T_i \in \\{0,1\\}$ is the treatment indicator and $X_i$ is a vector of covariates. The goal is to estimate and provide confidence intervals for the causal effect,

$$ \tau = E[Y_i \mid T_i = 1] - E[Y_i \mid T_i = 0].$$

The simplest estimator here is the difference in means estimator

$$ \hat{\tau}_{\text{diff. means}} = \text{average}(Y_i: T_i=1) - \text{average}(Y_i: T_i=0).$$

Confidence intervals can be constructed using a simple central limit theorem. One can do substantially better (shorter confidence intervals, better point estimator) by utilizing information in $X_i$ to reduce variance and there have been many approaches to do so. This paper proposes an approach that uses cross-fitting (splitting all $i$'s into $K$ folds) and arbitrary predictive models of $E[Y_i \mid X_i]$ (learned out-of-fold) and combines the results using a linear regression adjustment. The authors also show how to compute confidence intervals for the method and prove they are asymptotic valid even if inconsistent predictive models are used.  The asymptotic variance is never worse than the difference in means estimator. The authors also show substantial gains for the analysis of randomized controlled experiments at Facebook.

**Limitations And Societal Impact:**

There is no discussion about potentials for negative societal impact.

**Main Review:**


The paper is extremely well-written and is a pleasure to read. The motivation, the estimation strategy, the assumptions, and the theoretical results are presented very clearly. The authors have spent a lot of time polishing this paper. Also I liked the discussion about implementation issues at Facebook.


Nevertheless, I am a bit hesitant about this paper, for the following reason. This specific research area (improving estimates from randomized controlled trials using potentially high-dimensional adjustments) has been congested for a few years already. The authors do a thorough job of providing references to related work, but there are so many works that they still miss out and do not cite equally relevant works on the same problem. Some examples of such references that I am aware of are the following.

* Wu, E., & Gagnon-Bartsch, J. A. (2018). The loop estimator: Adjusting for covariates in randomized experiments. Evaluation review, 42(4), 458-488.

* Spiess, J. (2018). Optimal estimation when researcher and social preferences are misaligned. Job Market Paper.

* Opper, I. M. (2021). Improving Average Treatment Effect Estimates in Small-Scale Randomized Controlled Trials.

* Aronow, P. M., & Middleton, J. A. (2013). A class of unbiased estimators of the average treatment effect in randomized experiments. Journal of Causal Inference, 1(1), 135-154.


The authors explain in the related work section, what is new in their proposal compared to existing works. However, I believe that stronger practical and/or theoretical evidence would be required to demonstrate that the new estimator is not just a minor variation of all these existing works.

On the practical side, the new method is only compared against simple baselines (difference in means and CUPED). What about all the other estimators that use adjustments based on ML methods?

On the theory side, it appears to me that the assumptions are made that simplify the proofs as much as possible, and are not the ones that would make a strong case for the advantages of the new method.

1) The most important motivation for this work appears to be the fact that the method works well even if the predictive models are bad. For example, the authors write for existing methods that "This approach is less appealing in this context, as it would greatly restrict the kind of ML methods that could be used for the prediction step.", while their method "performs well in the presence of poor-quality predictions". The way poor predictions are defined in this work is as predictive functions $\hat{g}$ that converge in $L^4$ to a fixed function $g_0$,
$$ \int [ \hat{g} - g_0]^4 dP \stackrel{p}{\to} 0.$$
$g_0$ may be different than the true regression function, but is the same across all folds of cross-fitting. This is a stronger assumption that the authors let on ("This is a weak condition, easy to satisfy in applications") and I think the authors need to either weaken it, or spend more time on justifying it. Guo and Basse (2020), use the term "stable" for predictive models with a very similar property. Also to me, having the same inconsistent limit which is also the same across folds, suggests a heavily regularized procedure, such as Ridge regression. What about predictive models that never converge to something or never settle down (e.g., $1$-nearest neighbors regression or CART that is grown very deeply or neural nets with increasing number of layers as $N$ increases)?

2) Related to the above point, it seems to me that finite-sample theory (ala Berry–Esseen) that does not require that the models settle down asymptotically would be very elegant in the present setting.

3) Would it be possible to give a more concrete condition that implies the (uniformly) minimum eigenvalue condition on $Z(g)Z(g)\top$, using the specific structure of the studied model (randomized controlled trial)?

4) The model studied is a superpopulation model with $(Y_i, X_i, T_i)$ drawn as independent and identically distributed samples from a superpopulation distribution. My sense of the field studying randomized controlled trials is that this model is typically considered to be a starting point in which results are "easy" to prove. The ultimate goal for inferential validity (Athey and Imbens, 2017, The econometrics of randomized experiments) however often is to provide inferential guarantees treating the potential outcomes and covariates as fixed, and conducting inference using only the randomness in the treatment assignment mechanism. Such an approach is mentioned in many of the works studied in the related work section (such as Cohen and Fogarty (2020), Guo and Basse (2020). The fact that here a superpopulation model is studied, is an important limitation compared to these other recent works. Could the results also be generalized when only treatment assignment is random?




-> After reading the response, I have raised my score to 5 from 4. I particularly appreciate simplifying the assumption I mentioned in question 3) above; the simplification is elegant and interpretable. Also I appreciate the clarifications to some of my other questions. I still however am not convinced that the new method would outperform existing methods in practice (e.g., in the application at Facebook), such as the cross-fitted AIPW type estimator and other methods cited. The only argument in favor of MLRATE as far as I can tell is the fact that it is never worse (asymptotically) than difference-in-means. I am not convinced of the practical importance of this result (it seems that it would only be relevant if the ML engineers/data scientists fitting the ML model did a really bad job, which furthermore would be detectable by cross-validation).








**Time Spent Reviewing:**

7

---

> ### Author Response · Authors · 2021-08-09
> **Response to Review Two**
>
> We are very grateful to the reviewer for their thorough reading of our paper and extremely valuable feedback. We address the points raised in turn:
>
> **Related Literature.**
>
> The papers cited by the reviewer do indeed belong to the most relevant part of the literature: model-agnostic variance reduction using general ML methods (as opposed to, e.g. Bloniarz, Liu, Zhang, Sekhon, and Yu [2016]; Wager, Du, Taylor, and Tibshirani [2016]; Lei, L and Ding, P [2020], which either assume some kind of consistency for the true data generating process, or analyze special cases of ML methods, such as the lasso under ultrasparsity conditions). We thank you for giving us the opportunity to clarify the differences between our work and the cited works.
>
> The LOOP estimator of Wu & Gagnon-Bartsch is a special case of the estimator class considered in Aronow & Middleton. The latter paper does not give an expression for the estimator variance when the nuisance functions are estimated from the data, limiting its usefulness in practice. As the LOOP authors note, their contribution is to provide such a variance formula. The statistical procedure in Spiess, when restricting to unbiased estimators of the ATE, amounts to the LOOP estimator. Thus, of these three papers, we focus our comments on the LOOP estimator.
>
> LOOP uses sample splitting in a similar way as MLRATE does: both estimate potentially complex models on one split, and generate predictions from these models on the other split, and then use those predictions to produce lower variance estimates of the ATE. There are two major drawbacks to LOOP, however. First, it comes with a variance estimate, but no procedure for constructing confidence intervals or test statistics. Second, unlike MLRATE, the LOOP estimator variance has no formal non-inferiority guarantee relative to the difference-in-means estimator. These are real practical obstacles to adoption of LOOP in the setting of online experimentation we consider.
>
> Opper’s procedure is also based on the estimator class in Aronow and Middleton, and is very similar to LOOP, with largely the same strengths and weaknesses. Opper suggests constructing test statistics of the sharp null hypothesis of exactly zero treatment effect for all individuals, and constructing CIs based on this or somewhat less restrictive kinds of null. Randomization inference-based CIs require much stronger assumptions on treatment effects for validity (and more computation), or may only give coverage guarantees on estimands other than the ATE—e.g. the Caughey, Dafoe and Miratrix paper cited by Opper provides CIs for the maximum treatment effect, not the average.
>
> We thus focus on linear regression adjustment and the difference-in-means estimators in our comparisons, because the LOOP-style estimators do not come with CIs.
>
> **Numbered Comments.**
>
> **Comment 1.** Agreed, it is certainly true that some models may not even converge to the same limit across folds. Consistency results are nonetheless available for many common ML algorithms, including random forests (Athey, Tibshirani, and Wager 2019, and references therein), gradient boosted decision trees (Biao and Cadre 2017), deep feedforward neural nets (Farrell, Liang and Misra 2021), and regularized linear regression in some asymptotic regimes (Knight and Fu, 2000). Convergence to the same inconsistent limit may be expected, for example, in misspecified parametric models, and regularized linear regression in other asymptotic regimes. We should clarify that our claims to generality are not absolute however, and while our convergence assumption is easier to satisfy than assumptions elsewhere in the cross-fitting literature (and even more so than those in the non-cross-fitting literature, where Donsker conditions are invoked), the analyst should certainly apply care in ensuring it holds for their chosen algorithm.
>
> **Comment 2.** Thank you for this suggestion: a finite sample analysis would indeed be a very attractive path to explore. The case in which models do not converge at all in large samples may well be amenable to such an approach. It may provide a neat characterization of uncertainty in a setting where the usual asymptotic tools fail to apply.
>
> **Comment 3.** Yes indeed, and many thanks for prompting us to investigate this point! An equivalent characterization under our setting is $p \in (0,1)$ and $\inf_g Var(g(X)) > 0$. This assumption can be motivated with reference to the scientific question at hand: restricting attention only to adjustment functions which exhibit nontrivial variation with respect to the value of the covariate is unlikely to hurt the amount of variance reduction achieved.
>
> Below is an explanation on why the above conditions are equivalent to our assumption 1 iv). First, for any deterministic $g$, note that $P[Z(g)Z(g)^\top] = A \otimes B$, where $\otimes$ denotes the Kronecker product, and
> $$A = \begin{pmatrix}1 & Eg(X) \\\\ Eg(X) & Eg(X)^2 \end{pmatrix},$$
> $$B = \begin{pmatrix}1 & p \\\\ p & p \end{pmatrix}.$$
> Therefore, any eigenvalue of $P(Z(g)Z(g)^\top)$ is the product of an eigenvalue of $A$ and an eigenvalue of $B$. Given all eigenvalues of $A$ and $B$ are nonnegative and bounded, $\inf_{g\in \mathcal G}\lambda_{min}(P[Z(g)Z(g)^\top]) > 0$ is thus equivalent to $\inf_{g\in \mathcal G}\lambda_{min}(A) > 0$ and $\inf_{g\in \mathcal G}\lambda_{min}(B) > 0$. Some calculations show that $\inf_{g\in \mathcal G}\lambda_{min}(A) > 0$ is equivalent to $\inf_{g\in \mathcal G}Var(g(X)) > 0$. Finally, $\inf_{g\in \mathcal G}\lambda_{min}(B) > 0$ is equivalent to $p\in (0, 1)$. The claim follows.
>
> **Comment 4.**
> This is an excellent point, and the design-based analysis of this problem, where randomness only comes from the treatment assignment to a fixed, finite population, is a natural follow-on to this work. We note, however, that in the empirical application we study, there is indeed a superpopulation from which individuals are drawn: experiments are commonly run on 1% of the available user base, and the estimand of interest is the ATE in the *whole* population, not in the randomly selected sample. Thus in this case the approximation of an infinite superpopulation may well be closer to reality than the design-based paradigm, and the variance expressions from the latter analysis may be misleadingly small, as they do not account for the fact that we need to extrapolate to the rest of the superpopulation.

---

> ### Public Comment · ~Keyu_Nie1 · 2021-12-07
> **Is it similar to CUPAC as well?**
>
> Just include Door-dash's post for reference https://doordash.engineering/2020/06/08/improving-experimental-power-through-control-using-predictions-as-covariate-cupac/

---

> > ### Public Comment · ~Yongyi_Guo1 · 2021-12-08
> > **Yes! We have similar ideas on variance reduction**
> >
> > Hi Keyu,
> >
> > Thanks for your interest in our work!
> >
> > I read the post as well as the short notes in the following link:
> >
> > https://www.researchgate.net/profile/Yixin-Tang-5/publication/345698207_Control_Using_Predictions_as_Covariates_in_Switchback_Experiments/links/5fab109b458515078107aa8b/Control-Using-Predictions-as-Covariates-in-Switchback-Experiments.pdf
> >
> > I think the authors share a similar idea with us — to reduce variance by creating a variable (independent of treatment assignment) that is more `aligned’ with the post-experiment outcome variable. Instead of MLRATE, the authors seems to directly use the predictions from a trained ML model with $(X_i, Y_i)$ for variance reduction. It’s still not very clear to me how the authors provide an asymptotically valid confidence interval to deal with the dependencies among the ML predictions (e.g. using CRSE).
> >
> > Please do not hesitate to reach out to me if you have any further thoughts!

---

### Official Review · Reviewer_xSJv · 2021-07-18

**Rating:** 7
**Confidence:** 3

**Summary:**

This paper proposes what the authors call a machine learning regression-adjusted treatment effect estimator to reduce the variance when estimating the average treatment effect in randomized controlled trials, by using covariates correlated with the outcome but independent of the treatment. The algorithm first trains a machine learning model to predict the outcome from the covariates, and then runs a regression of the outcome on the treatment, predicted model and their cross-product, giving the regression coefficient of the treatment as the estimator. The authors prove consistency and asymptotic normality for their estimator under general conditions. As a main strength, the proposed estimator offers variance reduction when the machine learning prediction is good, and at the same time is robust to poor predictions - in the sense that its asymptotic behavior is no worse than the baseline difference-in-means estimator even if the prediction is bad.

The authors' main contributions are: 1) From the theoretical perspective, they propose an estimator with lower variance which is robust and easy to implement, and prove the consistency and asymptotic normality. 2) From the practical perspective, their estimator performs better than the baseline difference-in-means estimator and the univariate linear regression adjustment procedure in Deng et al. (2013) when applying to 48 outcome metrics commonly monitored in Facebook.

**Limitations And Societal Impact:**

Some limitations are discussed in the Main Review above.

**Main Review:**

Originality: The paper combines several existing elements to construct a method that has multiple advantages. The main novelty is this crafty combination. The individual concepts, on the other hand, are all viewed as known to an extent: The use of cross-fitting to avoid overfitting is widely known, and the main structure of their estimator which first estimates a nonlinear model (i.e., the machine learning step) and then calibrates it in a linear regression is motivated from Cohen and Fogarty (2020).

Quality: The proposed estimator appears advantageous on multiple fronts, all justified by theory. First it allows for general prediction models. Second, the estimator is easy to implement, including the confidence interval. Third, the estimator is robust which is important for practical use: If the prediction model is good, then the estimator could reduce the variance compared to the baseline estimator; if the prediction model is bad, the asymptotic behavior is no worse than the baseline.

Clarity: This paper is well organized. All the theoretical results are stated clearly and easy to read, and the authors also provide good intuition before presenting the technical details.

Significance: The proposed estimator appears significant, theoretically sound and easy to use, and the experimental results further support its practical merits.

Other Comments:

- In the statement of Corollary 1 and Proposition 3, the assumption $0<p<1$ in Proposition 2 seems to be also needed.

- In line 216 on page 6, I'm not sure how the quantity $\sigma^{2}/[\sigma_{Y_{C}}/(1-p)+\sigma_{Y_{T}}^{2}/p]$ is simplified as $1-Corr(Y,g_{0}(X))^{2}$ in the special case considered in this paragraph. The authors should clarify or correct it.

- At the end of Section 3, the authors say their algorithm applies equally to the high-dimensional regime, but without evidence. The authors may want to provide high-dimensional experimental results or tone down this claim.

- P.4, footnote: constant$\rightarrow $deterministic

**Time Spent Reviewing:**

3 hours

---

> ### Author Response · Authors · 2021-08-09
> **Response to Review One**
>
> We are very grateful to the reviewer for their excellent feedback and close reading of our paper. Below are our point-to-point responses to the comments raised in the "Other Comments" section.
>
> **Comment 1**.
> Thank you for your comment. Corollary 1 and Proposition 3 are both based on Assumption 1, which implicitly ensures that $0 < p < 1$. Specifically, Assumption 1 iv) ($\inf_g \lambda_{min}(P[Z(g)Z(g)^\top]) > 0$) implies $p \in (0, 1)$. We agree that the statement of this assumption is rather opaque however, and in our reply to reviewer 2, we show a much more straightforward and equivalent characterization of Assumption 1 iv), namely $p \in (0,1)$ and $\inf_g Var(g(X)) > 0$.
>
> **Comment 2**.
> Thank you for raising this point. As you may be alluding to, the $Y$ on the RHS is missing a subscript, and should be $Y_T$ or $Y_C$ (either is correct). We fully agree that further clarification is helpful for readers, and we should justify this fact in the appendix. The equivalence of the two forms can be justified as follows: Notice that $(\beta_{0, i})_\{i\in \{0, 1, 2, 3\}\}$ is the probabilistic limit of the OLS coefficient of $Y_i$ regressed on $(1, T_i, g_0(X_i), T_ig_0(X_i))$. This regression is equivalent to two separate OLS regressions on the treatment group and the control group:
>
> $$
> Y_i = (\beta_0 + \beta_1) + (\beta_2 + \beta_3) g_0(X_i) + \epsilon_i\quad \text{on treatment,}
> $$
> $$
> Y_i = \beta_0 + \beta_2 g_0(X_i) + \epsilon_i\quad \text{on control.}
> $$
>
> In our simplified case, $ \sigma_{Y_C} = \sigma_{Y_T}$ and $\beta_{0, 3} = 0$. We can write $\beta_{0, 2}$ as the probabilistic limit of the OLS slope estimates in both regressions:
> $$
> \beta_{0, 2} = Corr(Y_T, g_0(X))\cdot \frac{\sigma_{Y_T}}{\sigma_g}
> = Corr(Y_C, g_0(X))\cdot \frac{\sigma_{Y_C}}{\sigma_g}.
> $$
> Substituting the first equality above into eq. (10) in the paper gives
> $$
> \sigma^2 = \frac{\sigma_{Y_C}^2}{1-p} + \frac{\sigma_{Y_T}^2}{p}-\frac{\sigma_g^2}{p(1-p)}\cdot \beta_{0, 2}^2
> =(1-Corr(Y_T, g_0(X))^2)\cdot \left(\frac{\sigma_{Y_C}^2}{1-p} + \frac{\sigma_{Y_T}^2}{p}\right).
> $$
> This leads to our claim. We can also use the law of total variance to show that $Corr(Y_T, g_0(X))$ can be substituted by $Corr(Y, g_0(X))$ in the RHS above if the ATE is zero, or alternatively will be an excellent approximation if the ATE is very small relative to the baseline variation in the outcome, as is commonly the case with underpowered experiments in practice.
>
> **Comment 3.**
> Thank you for your comment. We also think it helpful to clarify how MLRATE does in the high dimensional case. Our explanations are below: First, our theory naturally applies to most high dimensional regimes. In fact, all the theoretical results only require that the estimated $\hat g_k$ converge to some $g_0$ in $L^4$. This condition allows the dimension of covariates to grow with $N$, and can be satisfied in a range of models and regimes, including high dimensional generalized linear models (e.g. Lasso, elastic net), nonparametric models (e.g. GBDT, random forest) and other ML models (e.g. neural networks). We present some related discussion and references in the response to review 2, below.
>
> Our simulations and real metric experiments are admittedly not high-dimensional settings in the sense of $N < dim(X)$. However, the dimension of the covariates reaches 100 in both, which is enough to encounter curse-of-dimensionality issues in nonparametric models without sufficient regularization, since $log(N) \ll dim(X)$. Thus, in addition to the theoretical guarantees, we view the existing results as at least providing some partial justification for the empirical applicability of MLRATE in high dimensional regimes.
>
> **Comment 4.**
> ’Deterministic’ is indeed a much better word than ‘constant’ for that footnote. Thanks for your suggestion!

---

### Decision · Program_Chairs · 2021-09-27

**Decision:**

Accept (Poster)

**Comment:**

The reviewers praise the quality of the write up (which seems to be a significant achievement wrt to the previous version of the paper) and the crafty combination of existing methods to build an estimator of causal effects. Both theoretical and empirical results are provided that back up the relevance of the contribution. This makes a paper on par with NeurIPS standard.
It is however expected for the authors to include in a subsequent version of the paper:
- an updated list of references provided by the reviewers and a discussion on how those works compare with the present work;
- empirical results based on competition models that are more elaborate than the ones proposed here: if the results of the method proposed in the present paper turn out not to be significantly better than those more elaborate models, then it is expected from the author to thoroughly share their insights on why this is the case.